# *Trust, But Verify:* A Self-Verification Approach to Reinforcement Learning with Verifiable Rewards

**Xiaoyuan Liu**[1,2*]   **Tian Liang**[2]   **Zhiwei He**[2,3]   **Jiahao Xu**[2]   **Wenxuan Wang**[4]
**Pinjia He**[1†]   **Zhaopeng Tu**[2†]   **Haitao Mi**[2]   **Dong Yu**[2]

[1]School of Data Science, The Chinese University of Hong Kong, Shenzhen
[2]Tencent   [3]Shanghai Jiao Tong University   [4]Renmin University of China

## Abstract

Large Language Models (LLMs) show great promise in complex reasoning, with Reinforcement Learning with Verifiable Rewards (RLVR) being a key enhancement strategy. However, a prevalent issue is "superficial self-reflection", where models fail to robustly verify their own outputs. We introduce RISE (**R**einforcing Reason**in**g with **S**elf-V**e**rification), a novel online RL framework designed to tackle this. RISE explicitly and simultaneously trains an LLM to improve both its problem-solving and self-verification abilities within a single, integrated RL process. The core mechanism involves leveraging verifiable rewards from an outcome verifier to provide on-the-fly feedback for both solution generation and self-verification tasks. In each iteration, the model generates solutions, then critiques its own on-policy generated solutions, with both trajectories contributing to the policy update. Extensive experiments on diverse mathematical reasoning benchmarks show that RISE consistently improves model's problem-solving accuracy while concurrently fostering strong self-verification skills. Our analyses highlight the advantages of online verification and the benefits of increased verification compute. Additionally, RISE models exhibit more frequent and accurate self-verification behaviors during reasoning. These advantages reinforce RISE as a flexible and effective path towards developing more robust and self-aware reasoners.

## 1   Introduction

Large Language Models (LLMs) have demonstrated remarkable potential in complex reasoning tasks. A promising avenue for further enhancing these capabilities is Reinforcement Learning (RL), particularly methods that utilize verifiable rewards (RLVR) from outcome verifiers [Gao et al., 2024, Guo et al., 2025, Lambert et al., 2024, Yue et al., 2025]. This paradigm, often applied to domains like mathematics where solution correctness can be programmatically evaluated, enabling models to improve through direct feedback on their generated solutions.

However, even with outcome-based RLVR, models may still learn to generate spurious reasoning without truly understanding the underlying logical process or developing robust self-assessment skills. This can lead to "superficial self-reflection" [Liu et al., 2025], where models struggle to reliably identify errors in their own reasoning and verify the correctness of their outputs, ultimately resulting in flawed solutions and suboptimal performance. While some approaches explicitly incorporate self-critique [Xi et al., 2024, Xie et al., 2025] to provide additional signals, the process of learning to solve problems and learning to verify solutions are often decoupled or lack direct, contemporaneous feedback for the verification skill itself within the RL loop.

---

[*] Work was done when Xiaoyuan and Zhiwei were interning at Tencent.
[†] Correspondence to: Pinjia He <hepinjia@cuhk.edu.cn> and Zhaopeng Tu <zptu@tencent.com>.

To address this limitation and foster more robust reasoning, we introduce **RISE** (**R**einforcing Reasoning with **S**elf-Verification) as a novel online reinforcement learning framework. RISE is designed to explicitly and simultaneously train an LLM to improve both its problem-solving ability and its capacity to verify its own generated solutions within a single, integrated RL process. The key idea is to leverage the verifiable reward signal from a rule-based outcome verifier not only to guide the generation of correct solutions but also to align the model's self-verification ability on-the-fly.

In the RISE framework, during each training iteration, the model first generates solutions for a batch of problems. Subsequently, using these on-policy generated solutions and the original problems, verification problems are constructed with a predefined template, prompting the model to critique its own solution and provide a score. The same outcome verifier used to assess problem solutions also provides ground-truth supervision for the verification task, based on an exact match between the predicted verification score and the ground-truth solution score. Both the problem-solving trajectories and the self-verification trajectories, along with their respective verifiable rewards, are then combined to update the model's parameters using a unified RL objective. This tight coupling enables the model to learn not only to solve problems, but also to critique and verify its own outputs, fostering a more dynamic and grounded self-improvement loop.

In our experiments, we implement and evaluate RISE using the Proximal Policy Optimization (PPO) algorithm, applying it to the 1.5B, 3B, and 7B base models from the Qwen2.5 series. Compared to a Zero-RL baseline, which incorporates only problem-solving supervision, RISE consistently improves reasoning accuracy and achieves up to a $2.8\times$ increase in verification accuracy on challenging mathematical benchmarks. Moreover, RISE outperforms instruction-tuned models across both tasks. For instance, RISE-3B improves the reasoning accuracy by 3.7 points and self-verification accuracy by 33.4 points compared with Qwen2.5-3B-Instruct.

We also find that this enhanced self-verification ability contributes to improved test-time performance. Specifically, RISE-3B and RISE-7B outperform standard majority voting by +0.2% and +1.9%, respectively, under a $k{=}4$ inference budget. Further analysis reveals that RISE enhances the internal reasoning process by encouraging more frequent and effective verification behaviors. Finally, our ablations demonstrate that online verification is crucial to the success of RISE.

Our main contributions are as follows:

- We introduce **RISE** (**R**einforcing Reasoning with **S**elf-Verification), a novel online reinforcement learning framework that explicitly and simultaneously trains LLMs to improve both problem-solving and self-verification capabilities within a single, integrated RL process, leveraging verifiable rewards for both tasks on-the-fly.

- We demonstrate, through extensive experiments on challenging mathematical reasoning benchmarks using a PPO-based implementation, that RISE significantly boosts problem-solving performance while instilling robust self-verification skills in the LLM.

- We provide comprehensive analyses elucidating the critical role of RISE's online verification mechanism, the benefits of scaling verification training compute, and how the developed self-verification capability contributes to more accurate and reliable solution generation.

## 2   Related Work

**RLVR for LLM Reasoning**    In the literature, reinforcement learning has been widely used to align language models with human preferences, typically through reward models or pairwise preference comparisons [Christiano et al., 2017, Ouyang et al., 2022, Rafailov et al., 2023]. More Recently, Reinforcement Learning with Verifiable Rewards (RLVR) has emerged as a powerful approach for improving the reasoning capabilities of LLMs in domains such as mathematics and programming [Jaech et al., 2024, Guo et al., 2025]. Using only outcome rewards, recent work has demonstrated the scalability of RL algorithms for LLM reasoning [Guo et al., 2025, Team et al., 2025, Zeng et al., 2025, Hu et al., 2025]. However, leveraging verifiable rewards not only for reasoning supervision but also as a direct training signal for self-verification remains underexplored, which is the main focus of RISE.

**Learning to Solve and Verify**    Solution generation and verification are two foundational capabilities of LLMs [Huang et al., 2024, Song et al., 2024], echoing the classic P versus NP dichotomy in

computer science [Wikipedia contributors, 2025]. In the context of LLM reasoning, previous work has focused on teaching models either to solve problems [Guo et al., 2025, Zelikman et al., 2022], to verify solutions [Wang et al., 2024, Lightman et al., 2023, Shi and Jin, 2025, Zhang et al., 2024], or to leverage the verification capability to perform self-improvement [Yuan et al., 2024, Xiong et al., 2025] and self-calibration [Huang et al., 2025]. More recently, Lin et al. [2025] introduced a self-play framework that trains LLMs to generate code and corresponding test cases through two-stage training, and Ma et al. [2025] proposed training methods that teach LLMs to self-verify and self-correct based on deliberately constructed trajectories. In contrast, we introduce an online RL framework that explicitly leverages verifiable reward signals to jointly align the model's problem-solving and self-verification abilities in a unified training process.

## 3  Reinforcement Learning Preliminaries

**Policy Gradient Methods**  The goal of RL is to learn a policy that maximizes the expected cumulative reward (namely return), denoted as the performance measure $J$. Policy gradient methods learn a parameterized policy that can select actions to maximize $J$ without consulting other value functions. Grounded by the *policy gradient theorem* [Sutton and Barto, 2018], the optimization is performed as gradient ascent based on the gradient of $J(\theta)$ with respect to the policy parameter $\theta$.

A large language model is naturally a parametrized policy $\pi_\theta$. The state at time $t$, denoted as $s_t$, is the concatenation of the prompt $\mathbf{x}$ and the response $\mathbf{y}_{<t}$ generated so far, while the action $a_t$ is the next token $y_t$. $T$ refers to total timestamps (response length + 1). Thus, the gradient can be expressed as:

$$\nabla_\theta J(\theta) = \mathbb{E}_{\mathbf{x}\sim\mathcal{D},\, \mathbf{y}\sim\pi_\theta} \left[ \sum_{t=0}^{T} \nabla_\theta \log \pi_\theta(y_t \mid \mathbf{x}, \mathbf{y}_{<t}) A_t \right].$$

The core part of this method is the advantage function $A_t$, which determines the extent to increase or decrease the probability of selecting this action (token) in the given state. In practice, the advantage function is implemented as cumulative discounted rewards subtracting an optional baseline, representing how much better an action is compared to the alternatives:

$$A_t = \sum_{t=t_0}^{T} \gamma^{t-t_0} r_t - b(s_{t_0}), \tag{1}$$

where $\gamma \in [0,1]$ is the discount factor for the future rewards and $r_t = R(s_t, a_t, s_{t+1})$ is the reward from the environment at time $t$. Different implementations of the baseline formulate multiple variants of policy gradient methods, including using learned state-value functions (e.g., REINFORCE [Williams, 1992], Actor-Critic [Barto et al., 1983]), group-level reward means (e.g., GRPO [Shao et al., 2024]), and leave-one-out (e.g. RLOO [Ahmadian et al., 2024]).

**Proximal Policy Optimization**  Proximal Policy Optimization (PPO) [Schulman et al., 2017] is a popular algorithm of Actor-Critic method, which incorporates a critic model $\phi$ to help estimate advantage for training the actor model $\theta$ (i.e., policy). One major improvement of PPO is penalizing excessive policy updates and thereby maintaining training stability. In practice, the objective of the actor model is defined as follows:

$$\mathcal{J}(\theta) = \mathbb{E}_t \left[ \min\left( r_t(\theta)\hat{A}_t, \text{clip}\left(r_t(\theta), 1-\epsilon, 1+\epsilon\right)\hat{A}_t \right) - \beta KL\left(\pi_\theta || \pi_{ref}\right) \right], \tag{2}$$

where $r_t(\theta) = \frac{\pi_\theta(y_t|x,y_{<t})}{\pi_{\theta_{old}}(y_t|x,y_{<t})}$. Clip() and KL() are two techniques used for limiting update magnitudes. With Generalized Advantage Estimation (GAE) [Schulman et al., 2015], the advantage is estimated as a $\lambda$-weighted sum of step-emporal-Difference (TD) errors:

$$\hat{A}_t = \delta_t + (\gamma\lambda)\delta_{t+1} + \cdots + (\gamma\lambda)^{T-t-1}\delta_{T-1}, \tag{3}$$

$$\text{where } \delta_t = r_t + \gamma V_\phi(s_{t+1}) - V_\phi(s_t).$$

$T$ denotes response length with token indexes from 0 to $T-1$. $V_\phi(s_t)$ is the value predicted by the critic model $\phi$ at state $s_t$, $r_t$ is the scalar reward from the environment at time $t$, and $\lambda \in [0,1]$ is the GAE parameter that trades off between bias and variance. In practice, we set $\lambda = \gamma = 1$, thus

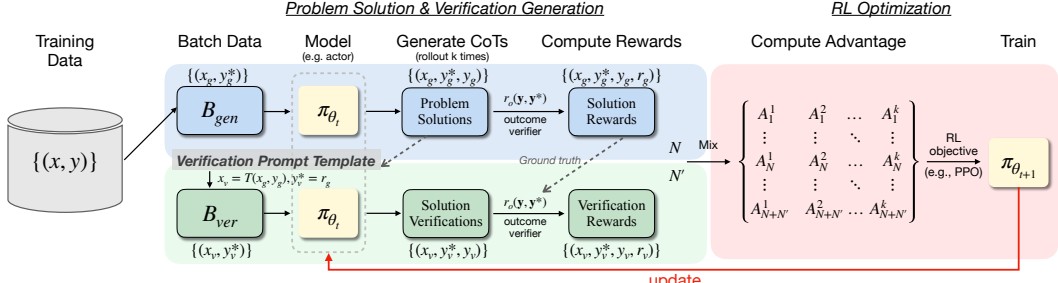

Figure 1: Illustration of RISE, which consists of two stages: (i) *Problem Solution & Verification Generation*: problems from the training batch are used to generate chain-of-thought solutions from the model. Problems and model solutions are then formatted as verification prompts to generate verifications of the solutions. (ii) *RL Optimization*: the original generation data and their verification are mixed as the new batch, and the model is optimized based on the RL objective.

making the per-token loss averaged over the full response length $T$. By design, $r_t = 0$ for $t < T - 1$, and $r_t = r$ for $t = T - 1$ (i.e., outcome reward). After we update the actor model, the critic model should also be updated for accurate value estimations. In practice, we use Mean Squared Error (MSE) to measure the prediction loss and perform the update:

$$\mathcal{J}(\phi) = \mathbb{E}_t \left[ \max \left( (V_\phi(s_t) - V_t^{targ})^2, \left( \text{clip}(V_\phi(s_t), V_{\phi_{\text{old}}}(s_t) - \epsilon, V_{\phi_{\text{old}}}(s_t) + \epsilon) - V_t^{targ} \right)^2 \right) \right],$$

$$\text{where } V_t^{targ} = V_{\phi_{\text{old}}}(s_t) + \hat{A}_t.$$

(4)

**Verifiable Reward** Unlike the rewards from conventional reward models which are continuous values denoting the goodness of the response, verifiable rewards are usually discrete signals representing the correctness of the final result [Lambert et al., 2024, Guo et al., 2025]. Given the prompt $\mathbf{x}$ and the complete response $\mathbf{y}$ from the LLM $\pi_\theta$, a classic verifiable reward is defined as a binary value produced by a deterministic outcome verifier $OV$: $r = OV(\mathbf{x}, \mathbf{y}) \in \{0, 1\}$, where $r = 1$ if and only if the final answer is exactly correct (e.g., the numeric result is mathematically equivalent to the ground truth answer) and $r = 0$ otherwise. In practice, an auxiliary format reward can be included to encourage the model to present its answer in a prescribed style.

## 4 Methodology: Reinforcing Reasoning with Self-Verification (RISE)

To address the challenge of superficial self-reflection, we propose RISE for self-improving reasoners, which is a scalable online RL method with explicit verification objective. **The key idea of RISE is the use of the verifiable reward signal from the rule-based outcome verifier to align the model's verification ability on-the-fly.** This enables us to teach the model to verify its own response at the same time it solves the problem, as depicted in Figure 1 and Algorithm 1.

### 4.1 Online Reasoning and Verification

**Problem Solution Generation** Given an initial model $\pi_\theta$ and a training set $D = \{(\mathbf{x_i}, \mathbf{y}_i^*)\}$ consisting of problems $\mathbf{x}_i$, and their corresponding ground-truth answers $\mathbf{y}_i^*$, we begin each RL iteration by sampling a data batch. At iteration $t$, the model first generates $k$ solutions for each problem in the batch, each comprising a chain-of-thought reasoning followed by a final answer.

Next, the reward is computed for each generated response. Following prior RLVR approaches, we define a rule-based outcome verifier (OV) that incorporates both answer and format correctness:

$$r_o(\mathbf{y}, \mathbf{y}^*) = \begin{cases} 1, & \text{boxed and matched} \\ -0.5, & \text{boxed but not matched} \\ -1, & \text{unboxed} \end{cases}$$

(5)

Here "matched" means the final answer in the generated solution $\mathbf{y}$ is mathematically identical to the provided ground truth $\mathbf{y}^*$, and "boxed" means the final answer in $\mathbf{y}$ is wrapped in the \boxed{}.

**Algorithm 1** RISE (PPO)

---

**Input** Language model $\pi_{\theta_{\text{init}}}$; outcome verifier OV; dataset $\mathcal{D}$; rollout number $K$; generation batch size $\mathcal{B}_g$, verification batch size $\mathcal{B}_v$; verification prompt template $\mathcal{T}$; total iteration $N$.

1: **Initialize:** actor $\pi_\theta \leftarrow \pi_{\theta_{\text{init}}}$, old-actor $\pi_{\theta_{\text{old}}}$, critic $\pi_\phi$, reference $\pi_{\text{ref}}$
2: **for** iteration $= 1$ **to** $N$ **do**
3:      Sample $\mathcal{B}_g$ samples for generation $\mathcal{P}_g = \{(\mathbf{x}_i, \mathbf{y}_i^*)\}_{i=1}^{\mathcal{B}_g} \sim \mathcal{D}$
4:      Get generation batch:                                ▷ Generate solutions
         $\mathcal{G} \leftarrow \left\{ (\mathbf{x}_i, \mathbf{y}_i^{(k)}, r_{\text{ov}}(\mathbf{y}_i^{(k)}, \mathbf{y}_i^*)) \mid \mathbf{y}_i^{(k)} \sim \pi_\theta(\cdot|\mathbf{x}_i), \, i \leq \mathcal{B}_g, \, k \leq K \right\}$
5:      Select $\mathcal{B}_v$ triples $\mathcal{P}' = \{(\mathbf{x}_i, \mathbf{y}_i, r_i)\}_{i=1}^{\mathcal{B}_v} \subseteq \mathcal{G}$ for verification
6:      $\mathcal{P}_v \leftarrow \left\{ (\mathcal{T}(\mathbf{x}, \mathbf{y}), r) \mid (\mathbf{x}, \mathbf{y}, r) \in \mathcal{P}' \right\}$    // each element is a new prob-ans tuple $(\mathbf{x}, \mathbf{y}^*)$
7:      Get verification batch:                                 ▷ Verify generations
         $\mathcal{V} \leftarrow \left\{ (\mathbf{x}_j, \mathbf{y}_j^{(k)}, r_{\text{ov}}(\mathbf{y}_j^{(k)}, \mathbf{y}_j^*)) \mid \mathbf{y}_j^{(k)} \sim \pi_\theta(\cdot|\mathbf{x}_j), \, j \leq \mathcal{B}_v, \, k \leq K \right\}$
8:      Get complete training batch $\mathcal{B} \leftarrow \mathcal{G} \cup \mathcal{V}$
9:      Estimate advantages $\hat{A}$ using Eq. (3)                         ▷ Joint optimization
10:     Update critic $\pi_\phi$ by critic loss in Eq. (4)
11:     Update actor $\pi_\theta$ by actor loss in Eq. (2); update $\theta_{\text{old}} \leftarrow \theta$
12: **end for**

**Output** Optimized actor model $\pi_\theta$

---

This produces the generation batch $\mathcal{G} = (\mathbf{x}, \mathbf{y}, r)$, where each element includes the input problem, a model-generated solution, and its associated reward.

**Online Solution Verification**    To construct verification data, we apply a predefined prompt template (see Figure 8) to $\mathcal{G}$, formatting the problem-solution pair into a new verification prompt $\mathbf{x}_{\text{ver}}$ that explicitly states the verification criteria and asks the model to critique the provided solution and assign a score. Since the criteria specified in the prompt are exactly the rules employed by the outcome verifier, the original reward $r$ from the generation phase is reused as the ground-truth score for the verification task. Thus, for each triple $(\mathbf{x}, \mathbf{y}, r) \in \mathcal{G}$, we construct the verification data as $(\mathbf{x}_{\text{ver}} = \mathcal{T}(\mathbf{x}, \mathbf{y}), \, \mathbf{y}_{\text{ver}}^* = r)$. In practice, the amount of verification data is controlled by the verification batch size, allowing a flexible balance between problem-solving and solution verification.

For each verification prompt, the model generates $K$ responses, each containing a natural language critique and a final score. These responses are evaluated using the same OV criteria as Eq. (5). Concretely, the verification reward is determined by comparing the score extracted from the model's verification response with the reward assigned by the rule-based outcome verifier for the same judged solution. A reward of $+1.0$ is given when the model correctly predicts the score in the specified format (e.g., \boxed{1}); a reward of $-0.5$ is assigned when the predicted score is incorrect but the format is valid; and a reward of $-1.0$ is applied when the response is in an invalid format (missing the \boxed{} wrapper). This process yields the verification batch $\mathcal{V} = \{(\mathbf{x}, \mathbf{y}, r)\}$, maintaining the same structure as the problem-solution batch.

## 4.2 RL Integration

The preceding *Online Reasoning and Verification* stage is architecturally agnostic to the choice of the underlying policy-gradient algorithm; its only algorithm-specific interface is the advantage estimator $\hat{A}$ used in the policy update. In our formulation, advantage values are computed from a concatenated mini-batch $\mathcal{B} = \mathcal{G} \cup \mathcal{V}$, encompassing samples from both the reasoning and verification tasks. Since every sample in $\mathcal{B}$ is annotated with a scalar reward and the action log-probability under the current policy $\pi_\theta$, any estimator that maps a sequence of state–action–reward tuples to an advantage can be incorporated without further structural change.

For our main experiments with PPO (see Algorithm 1), we apply GAE (Eq. 3) independently to each trajectory. The generation and verification trajectories are jointly processed within the same stochastic gradient descent (SGD) step, enabling the actor to be optimized with respect to both types of data. Meanwhile, the shared critic learns a unified value function across tasks. PPO's clipping mechanism further ensures that updates remain stable within a consistent trust region.

Table 1: Detailed results of RISE and other baseline methods on various math benchmarks. Zero-RL models are trained under the same setting as RISE, but without the verification objective.

| Model | Reasoning | | | | | | Self-Verification | | | | | |
|---|---|---|---|---|---|---|---|---|---|---|---|---|
| | MATH | AIME | AMC | Mine. | Olym. | Avg. | MATH | AIME | AMC | Mine. | Olym. | Avg. |
| GPT-4o | 79.0 | 13.3 | 55.0 | 50.0 | 42.5 | 48.0 | 83.4 | 33.3 | 67.5 | 50.4 | 54.4 | 57.8 |
| *Qwen2.5-1.5B* | | | | | | | | | | | | |
| Base | 2.0 | 0.0 | 1.9 | 0.8 | 0.6 | 1.1 | 19.4 | 21.9 | 22.7 | 15.9 | 21.1 | 20.2 |
| Instruct | 37.5 | 0.8 | 19.4 | 8.3 | 11.7 | 15.5 | 48.8 | 22.1 | 36.5 | 36.9 | 29.6 | 34.8 |
| SFT | 10.1 | 0.0 | 4.1 | 1.8 | 2.0 | 3.6 | 19.0 | 5.8 | 12.3 | 10.5 | 10.9 | 11.7 |
| Zero-RL | **55.3** | 2.1 | 25.9 | **17.4** | 19.5 | 24.0 | 54.1 | 5.0 | 30.7 | 21.0 | 23.0 | 26.8 |
| **RISE** | 54.6 | **2.9** | **27.5** | 17.2 | **19.8** | **24.4** | 75.9 | 85.0 | 70.6 | 66.0 | 74.9 | 74.5 |
| *Qwen2.5-3B* | | | | | | | | | | | | |
| Base | 32.7 | 1.3 | 15.3 | 10.3 | 10.7 | 14.1 | 39.5 | 13.6 | 22.5 | 29.9 | 21.2 | 25.3 |
| Instruct | 61.0 | 3.8 | 34.1 | 25.6 | 24.6 | 29.8 | 65.6 | 21.0 | 45.5 | 37.6 | 35.0 | 40.9 |
| SFT | 14.4 | 0.4 | 5.3 | 2.9 | 2.8 | 5.2 | 21.5 | 2.1 | 10.9 | 17.9 | 13.2 | 13.1 |
| Zero-RL | 64.2 | 6.7 | 37.5 | **27.4** | **26.6** | 32.5 | 64.9 | 13.0 | 39.7 | 30.3 | 31.2 | 35.8 |
| **RISE** | **64.3** | **7.9** | **42.5** | 26.2 | **26.6** | **33.5** | 81.0 | 86.3 | 74.4 | 56.1 | 73.6 | 74.3 |
| *Qwen2.5-7B* | | | | | | | | | | | | |
| Base | 38.3 | 2.1 | 21.9 | 11.9 | 13.2 | 17.5 | 58.4 | 45.9 | 51.5 | 48.4 | 48.4 | 50.5 |
| Instruct | 73.8 | 10.0 | 50.6 | **35.9** | 35.8 | 41.2 | 77.2 | 26.3 | 57.0 | 40.2 | 45.2 | 49.2 |
| SFT | 28.7 | 0.8 | 13.8 | 6.2 | 7.2 | 11.3 | 40.5 | 36.6 | 47.4 | 39.2 | 36.1 | 40.0 |
| Zero-RL | 74.5 | 12.1 | 51.3 | 34.2 | **36.7** | 41.7 | 75.9 | 21.7 | 56.5 | 37.3 | 41.6 | 46.6 |
| **RISE** | **74.8** | **12.5** | **55.9** | 34.6 | **36.7** | **42.9** | 83.8 | 75.0 | 72.5 | 48.6 | 65.9 | 69.2 |

## 5 Experiment

### 5.1 Experiment Setup

**Dataset** We follow the previous study [Zeng et al., 2025] to utilize MATH-Hard (Level 3–5) [Hendrycks et al., 2021] as our training set, which in total comprising 8,523 problems. This training set is used for all SFT baselines, Zero-RL baselines, and RISE models.

**Models** We conduct our main experiments on three Qwen2.5 models [Yang et al., 2024] with different sizes (i.e., 1.5B, 3B, and 7B) for their strong reasoning capabilities. The RL training of our models is based on the verl [Sheng et al., 2025] framework with a train batch size of 1024 and a mini-batch size of 128. We follow [Zeng et al., 2025] by setting the sampling temperature to 1.0 and rollout 8 responses for each problem. The RISE models have a default verification batch size 128. We set the RL configurations same for RISE models and Zero-RL models, ensuring a fair comparison. Additionally, we include the RL experiments on Qwen3 models in Appendix H.1, where we observe larger performance gains, further demonstrating the effectiveness of RISE on newer models.

**Benchmarks** We evaluate model performance on standard mathematical reasoning benchmarks: MATH500 [Hendrycks et al., 2021, Lightman et al., 2023], Minerva Math [Lewkowycz et al., 2022], OlympiadBench [He et al., 2024], and competition-level benchmarks AIME 2024 and AMC 2023. Following [Zeng et al., 2025], we generate 8 responses per problem using a sampling temperature of 1.0, and report Pass@1 accuracy [Chen et al., 2021] as the evaluation metric. Solution correctness is based on exact match of the final answer, and verification correctness depends on the agreement between the predicted verification score and the score from the outcome verifier.

### 5.2 Experimental Results

Table 1 presents the results of RISE across model sizes and benchmarks.

**RISE significantly enhances self-verification capabilities while improving reasoning performance.** RISE models consistently outperform their Zero-RL counterparts across both reasoning and self-verification tasks on all model sizes. The improvement in self-verification is particularly dra-

matic: RISE-1.5B achieves 74.5% average verification accuracy compared to just 26.8% for Zero-RL, representing a 47.7 percentage point improvement. This demonstrates that our integrated approach successfully develops robust self-verification skills while simultaneously enhancing problem-solving capabilities. Notably, **the verification improvements are particularly pronounced on the most challenging benchmarks** like AIME24 and OlympiadBench, suggesting that RISE enables models to better recognize their limitations and errors on difficult problems.

**Scaling model size improves reasoning performance while maintaining strong verification capabilities.** Scaling model size from 1.5B to 7B parameters consistently enhances reasoning performance across all benchmarks. Interestingly, the verification performance of RISE models remains consistently high across model sizes, with all models achieving over 69% average accuracy. The ability to maintain strong verification capabilities while scaling reasoning performance aligns with our contribution of developing a framework that simultaneously improves both critical capabilities.

**RISE models outperform standard SFT and base models by a substantial margin.** The results clearly demonstrate that RISE models substantially outperform their SFT and base model counterparts. For instance, RISE-7B achieves 42.9% average reasoning accuracy compared to just 11.3% for SFT-7B and 17.5% for the base model. This substantial improvement demonstrates the effectiveness of our integrated, online learning approach built upon state-of-the-art RLVR paradigms.

### 5.3 Test-Time Scaling with Self-verification

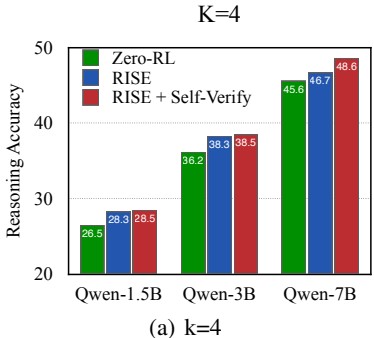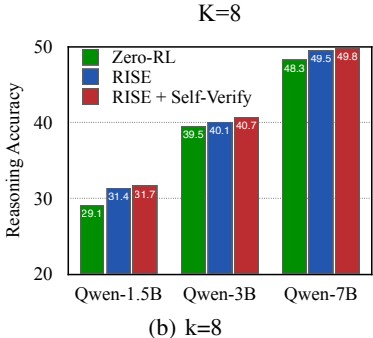

Figure 2: Test-time scaling performance across different sampling budgets ("k").

To further evaluate the benefits of the enhanced self-verification capabilities developed by RISE, we investigate its impact at test-time using self-consistency majority voting ("maj@k") [Wang et al., 2023] and verification-weighted majority voting. In the latter, following [Wang et al., 2024], the model's self-generated verification scores for each candidate solution are used to weight its contribution in the majority vote. The results, presented in Figure 2, compare RISE models against Zero-RL models across different sampling budgets ("k=4" and "k=8"). With the lightweight verification cost (Appendix G.2), the experiments maintain fairness in terms of total sampling cost.

**RISE consistently improves test-time scaling performance with self-verification and majority voting.** RISE models outperform their Zero-RL counterparts when employing test-time strategies such as majority voting and verification-weighted selection. Across model sizes and sampling budgets, RISE achieves higher average accuracy, with the largest relative gains observed when self-verification scores are used to re-rank majority votes. For example, RISE-7B achieves an average score of 49.8% with $k = 8$ + self-verify, surpassing Zero-RL's 48.3% under the same conditions. This consistent improvement confirms the effectiveness of integrating self-verification in both training and inference.

**Verification-weighted voting delivers further accuracy gains.** Incorporating self-verification scores as weights in the voting process leads to additional accuracy improvements for all RISE models. For instance, RISE-3B and RISE-7B models see improvements of +0.2% and +1.9% over standard majority voting at the $k = 4$ budget, respectively. These results indicate that the self-verification policy learned by RISE provides meaningful confidence signals for answer calibration.

## 5.4 Comparison with Off-the-shelf Verifiers

We further compare the verification accuracy between our RISE models as self-verifiers and off-the-shelf verifiers, including a discriminative verifier (Math-Shepherd-7B [Wang et al., 2024]) and a generative verifier (GPT-4o [OpenAI, 2024]). Specifically, we use the verification prompt in Figure 8 for both RISE models and GPT-4o and adhere to the original logic for Math-Shepherd to verify the generated solutions. The results of RISE-1.5B, 3B and 7B are presented in Figure 3, which show that RISE models consistently outperform existing outcome verifiers in judge their solutions' correctness. This serves as a great advantage for the model to further improve its test-time performance, by leveraging the self-verification signal either externally or internally. Detailed results and evaluation implementation can be found in Appendix D.

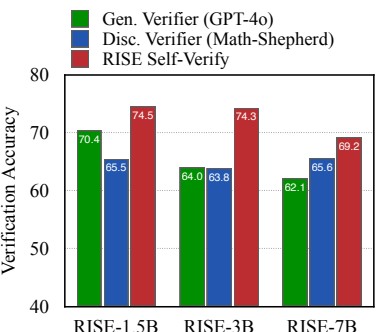

Figure 3: Comparisons between RISE (self-verify) and off-the-shelf verifiers.

## 5.5 Analysis

In this section, we provide some insights into how RISE improves performance.

**RISE demonstrates robust and simultaneous learning of problem-solving and self-verification, with self-verification skills developing notably faster across different model scales.** The learning curves, illustrated by the reward trends in Figure 4, reveal a consistent and steady improvement in both reasoning (problem-solving) and self-verification rewards throughout the RL training process for all evaluated models. This uniform positive progression across varying model sizes highlights the robustness of the RISE framework in co-training

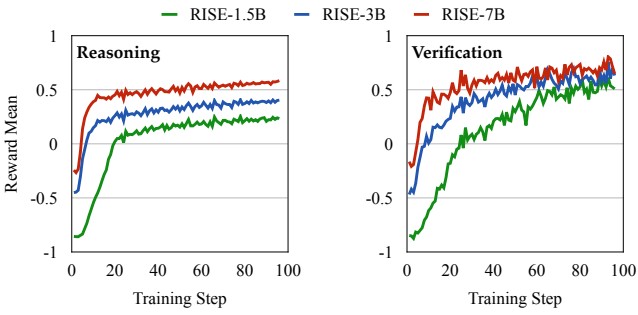

Figure 4: Reasoning and verification reward at train time.

these two abilities, a core contribution of our work. A key observation is that the self-verification reward generally exhibits a more rapid increase and reaches a higher relative level compared to the problem-solving reward within the same training period. This aligns with the "Generation-Verification Gap" posited by Song et al. [2024], suggesting that models might acquire verification capabilities more readily than problem-solving abilities.

**Impact of Verification Compute** In the main experiment, we trained our RISE models with a verification batch size of 128, which is 12.5% of the generation batch 1024. We further explore the model performance by scaling up the verification data batch, i.e., the train-time compute, up to 100% of the generation batch. In practice, we choose the percentages S of $\{0, 12.5\%, 25\%, 50\%, 100\%\}$ and perform experiment on our RISE models. The results are shown in Figure 5. The problem-solving perfor-

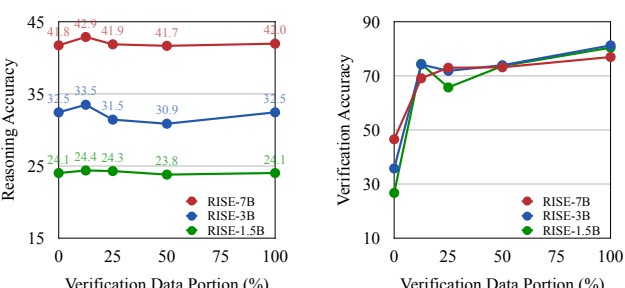

Figure 5: Impact of verification data ratio.

mance first increases, and then slightly decreases, and finally increases across the benchmark, maintaining at a high level. Furthermore, the verification performance keeps scaling with more training compute, indicating the robustness of scalability of our RISE method.

**Online and Offline Verification** We validate the effectiveness of online verification by comparing it to a offline variant, where the verification data are collected from a distant policy and directly added to the training set. In practice, we select the policy at step 96 (final step) of the Zero-RL model and use its generated solutions to construct offline verification set. In the experiment, we keep the portion of verification data and the training batch size same to eliminate other influence factors, making the only changing variable the source of the verification data. Figure 6 shows the results. While the problem-

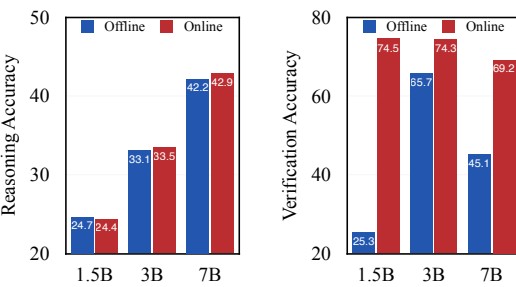

Figure 6: Comparisons between online and offline verification.

solving performance of offline verification models are on par with the online ones, they have a significant drop in terms of self-verification accuracy, which indicates the importance of online verification designed in our RISE method.

**Enhanced Verification for Reasoning** Besides leveraging the self-verification ability externally during the test-time as in § 5.3, such ability is also internalized by the model to enhance its reasoning generation process. To analysis this effect from the quantitative perspective, we measure the *Verification Frequency* and *Self-Verified Reasoning Accuracy* in model's problem-solving process. Inspired by [Yeo et al., 2025], we use a set of verification keywords to select the responses containing self-verification behaviors, namely {"verify", "verifying", "recheck", "validate", "re-evaluate"}.

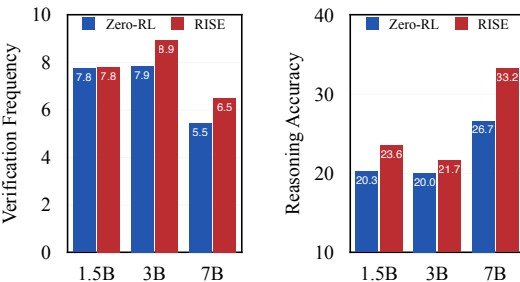

Figure 7: Verification frequency (left panel) and its effectiveness in reasoning (right panel).

Figure 7 (left) presents the first result, where the proportion of responses containing explicit verification phrases consistently rises after RISE training. The increase is modest for the 1.5 B model (+0.05 absolute) but becomes more pronounced as model scale grows (+1.09 for 3 B and +1.05 for 7 B). Since the decoding hyper-parameters are fixed, the higher verification frequency arises from policy updates that jointly optimize for both solution and verification rewards, an effect enabled by our first contribution, which integrates verification into a unified RL training loop rather than treating it as a post-hoc heuristic. This mechanism allows models to check their intermediate solutions before reaching the final conclusion in more cases.

Table 2: Examples of verification behaviors of different methods on Qwen2.5-7B: RISE can incur more effectiveness self-verification in the reasoning.

| Problem | What is the least positive integer multiple of 30 that can be written with only the digits 0 and 2? |
|---|---|
| Zero-RL | To find the least positive integer multiple of 30 that can be written with only the digits 0 and 2, we need to consider the properties of multiples of 30 . . . Since 222 is divisible by both 2 and 3, it is a multiple of 30. **Let's verify:** 222 is even, so it is divisible by 2. The sum of the digits of 222 is $2 + 2 + 2 = 6$, which is divisible by 3. Thus, 222 is indeed a multiple of 30. Therefore, the least positive integer multiple of 30 that can be written with only the digits 0 and 2 is $\boxed{222}$ . |
| RISE | To find the least positive integer multiple of 30 that can be written with only the digits 0 and 2, we need to . . . Therefore, the smallest number $N$ is 2220. **To verify:** 2220 ends in 0, so it is a multiple of 10. The sum of the digits of 2220 is $2 + 2 + 2 + 0 = 6$, which is a multiple of 3. Since 2220 satisfies both conditions, it is the least positive integer multiple of 30 that can be written with only the digits 0 and 2. Thus, the final answer is $\boxed{2220}$ . |

Simply asking the model to "verify" is not sufficient; the verification needs to be correct. In Figure 7 (right), we measure the accuracy of problems on which the model self-verifies. RISE yields clear improvements over Zero-RL for self-verified reasoning accuracy at every scale: +3.3% (1.5 B), +1.7% (3 B), and a striking +6.5% (7 B). These gains show that RISE's online verifier reward shapes the policy toward not only producing more verifications, but also ones that align with ground truth.

The case in Table 2 illustrates this distinction. Zero-RL "verifies" 222 by merely restating divisibility rules, overlooking the necessity of a trailing zero for multiples of 30. RISE, in contrast, recomputes both the units-digit constraint and the digit-sum divisibility test, ultimately validating the answer of 2220. Such structured, multi-step verification reflects a genuinely internalized skill and explains the quantitative trend that higher verification frequency correlates with higher reasoning accuracy.

# 6 Conclusion

In this work, we introduced RISE, a novel online reinforcement learning framework that integrates problem-solving with explicit self-verification training for LLMs. By leveraging verifiable rewards for both generation and verification tasks within a unified RL objective, RISE aims to overcome superficial self-reflection and foster more robust reasoning capabilities. Our experiments, primarily using PPO with Qwen2.5 models on diverse mathematical reasoning benchmarks, demonstrate that RISE significantly improves problem-solving accuracy while concurrently developing strong self-verification skills. Notably, RISE models learn to verify their own solutions more effectively than off-the-shelf verifiers and further benefit from leveraging this capability at test time. Together, RISE provides a promising direction for building more reliable and self-aware LLM reasoners, adaptable to various policy-gradient algorithms and extendable to other domains with verifiable rewards.

## Acknowledgment

This paper was supported by the Guangdong Basic and Applied Basic Research Foundation (No. 2024A1515010145) and the Shenzhen Science and Technology Program (Shenzhen Key Laboratory Grant No. ZDSYS20230626091302006).

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

# A Limitations

On experiments: First, after training the LLM to self-verify its outputs, it may act as a pseudo–rule-based verifier to further guide RL training without relying on ground-truth labels. This opens the possibility of self-improvement on unlabeled data, which we leave for future work as it lies beyond the main scope of this study. Second, the training data is exclusively drawn from math reasoning tasks. Given the observed generalization performance, we expect the effectiveness of RISE to persist when trained on other domains with verifiable rewards. Third, the metrics adopted for evaluating verification effectiveness have certain limitations, as they do not capture the necessity of the emerged verification behaviors. In this work, we use final reasoning accuracy as an indicator of goodness, while future work could develop more fine-grained criteria that better quantify the necessity such as partial trajectory correctness or model uncertainty.

On algorithm: RISE trains the LLM as a generative verifier that produces natural language critiques, which has shown benefits for both reasoning and test-time scaling. An alternative design is to co-train a discriminative verifier with a separate classification head. While it remains unclear how RISE would perform in that setting, we believe this does not affect our main contributions which demonstrate the effectiveness of generative verification in improving problem-solving capabilities.

# B Prompt Templates

---

**Prompt Template**

Below you are presented with a question and a tentative response. Your task is to evaluate and assign a rating to the response based on the following clear criteria:

Rating Criteria:

1. Missing final answer enclosed in \\boxed{} at the end: assign \\boxed{-1}.
2. Correct response with the final answer enclosed in \\boxed{} at the end: assign \\boxed{1}.
3. Incorrect response with the final answer enclosed in \\boxed at the end: assign \\boxed{-0.5}.

### Question Begin ###
{**Question**}
### Question End ###

### Response Begin ###
{**Response**}
### Response End ###

Briefly summarize your analysis, then clearly state your final rating value enclosed in \\boxed{} at the end.

---

Figure 8: Verification prompt used in the experiment.

---

**Prompt Template**

<|im_start|>system
Please reason step by step, and put your final answer within \\boxed{}.<|im_end|>
<|im_start|>user
{**Input**}<|im_end|>
<|im_start|>assistant

---

Figure 9: Prompt template used in the training and evaluation.

Figure 10: Prompt template used for Qwen base model evaluation.

## C   Training Details

During RL training, we set the actor's clipping ratio to 0.2 and disable the KL penalty loss. The critic uses a clipping range of 0.5. The learning rates are fixed at $5 \times 10^{-7}$ for the actor and $9 \times 10^{-6}$ for the critic. The KL divergence coefficient is set to $1 \times 10^{-2}$. We limit the maximum response length to 3000 tokens, which already results in a negligible clip ratio. The full dataset is trained for 12 epochs. This configuration is shared across both the Zero-RL and RISE models.

For the SFT baseline models, we use a batch size of 32 and apply a cosine learning rate scheduler with a learning rate of $2 \times 10^{-5}$ and a warm-up ratio of $1 \times 10^{-3}$. The dataset is trained for 3 epochs.

## D   Evaluation Details

### D.1   Verification Evaluation with Other Verifiers

To evaluate the verification accuracy of RISE and GPT-4o (prompted as a verifier), we extract the final verification score from each response and normalize it to either +1 (predicted correct) or 0 (predicted incorrect). The normalization is defined as:

$$s_{\text{normalized}} = \begin{cases} 1, & s = 1 \\ 0, & \text{otherwise} \end{cases}$$

which aligns with the criteria used by the rule-based outcome verifier. For the Math-Shepherd model, which outputs a continuous score in the range $[0, 1]$ (with 0 indicating the solution/step is predicted to be incorrect and 1 indicating correct), we apply a threshold of 0.5 for normalization:

$$s_{\text{normalized}} = \begin{cases} 1, & s > 0.5 \\ 0, & \text{otherwise} \end{cases}$$

After normalization, we compute verification accuracy by directly comparing the predicted scores against those returned by the outcome verifier.

### D.2   Weighted Majority Voting with Self-Verification

In § 5.3, we explore the combination of self-consistency and self-verification in test time following [Wang et al., 2024]. In practice, we initially classify solutions into distinct groups according to their final answers. Following that, we extract and normalize the self-verification scores and normalize them as +1 (correct) and 0 (incorrect) as in D.1. Since our score are binary and could lead to an unexpected zero sum, we integrate Laplace smoothing for computing the mean score for the answer. Formally, the final selected answer based on $N$ candidate solutions is:

$$a_{\text{maj@N+self-verify}} = \text{argmax}_a \underbrace{\sum_{i=1}^{N} \mathbb{I}(a_i = a)}_{\text{frequency}} \cdot \underbrace{\frac{\alpha + \sum_{i=1}^{N} S(p, s_i)}{N + \alpha d}}_{\text{smoothed mean score}}. \tag{6}$$

where $S(p, S_i)$ is the score of the $i$-th solution assigned by the self-verification. In practice, we set $\alpha = 2$ and $d = 2$ empirically, suggesting a prior belief of a 0.5 average score.

# E Licenses

**Datasets and Benchmarks.** The training dataset is derived from MATH (MIT License). We evaluate on five benchmarks: MATH 500 (MIT License), AIME 2024 (CC0: Public Domain), AMC 2023 (Apache License 2.0), Minerva Math (license not found), and Olympiad Bench (MIT License).

**Framework.** RL training is based on verl v0.2 (Apache-2.0 license), and SFT training is based on trl [von Werra et al., 2020] v0.14.0 (Apache-2.0 license). Evaluation is performed using vllm framework [Kwon et al., 2023] v0.7.2 (Apache-2.0 License) and the script is based on OpenMathInstruct-2 [Toshniwal et al., 2024].

**Models.** We train our models based on the Qwen2.5 series. Specifically, Qwen2.5-1.5B[3] and Qwen2.5-7B[4] are released under the Apache License 2.0, while Qwen2.5-3B[5] is released under a custom Qwen Research license. We also compare against Math-Shepherd[6] model (license not found), and GPT-4o (accessed via OpenAI API, governed by OpenAI Terms of Use[7]).

# F Computing Resources

All experiments were conducted on machines with AMD EPYC 9K84 96-core CPUs and 8 NVIDIA H20 GPUs. Our code is primarily based on Python 3.12.2 and PyTorch 2.5.1. RL training took approximately 1 day for Qwen-1.5B and Qwen-3B, and 2 days for Qwen-7B. The SFT baseline training required about 2 hours per model, and evaluation took roughly 1 hour per model. In total, the experiments consumed approximately 700 GPU hours. Including preliminary and failed runs, the overall project required more compute.

# G Detailed Experiment Results

## G.1 Detailed Comparison with off-the-shelf verifiers

In § 5.4, we report the average verification accuracy across the five benchmarks. Here, we present the detailed verification accuracy comparison between RISE models, Math-Shepherd, and GPT-4o on each evaluation benchmark.

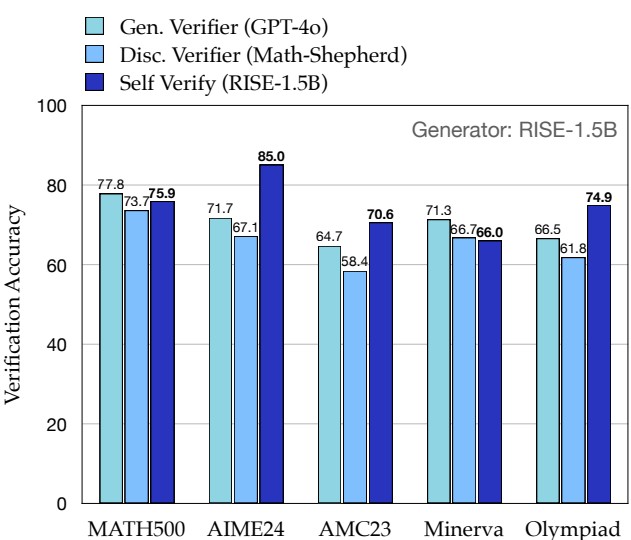

Figure 11: Detailed comparisons of verification accuracy between RISE-1.5B and other verifiers.

---

[3] https://huggingface.co/Qwen/Qwen2.5-1.5B

[4] https://huggingface.co/Qwen/Qwen2.5-7B

[5] https://huggingface.co/Qwen/Qwen2.5-3B

[6] https://huggingface.co/peiyi9979/math-shepherd-mistral-7b-prm

[7] https://openai.com/policies/row-terms-of-use/

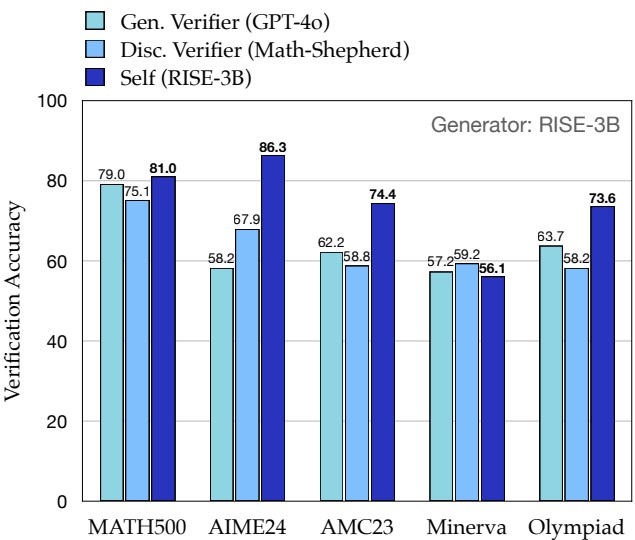

Figure 12: Detailed comparisons of verification accuracy between RISE-3B and other verifiers.

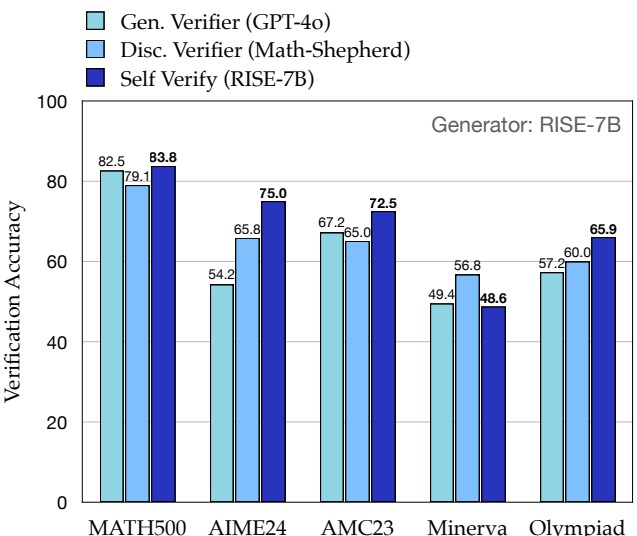

Figure 13: Detailed comparisons of verification accuracy between RISE-7B and other verifiers.

## G.2 Verification Cost

To assess the verification cost, we compute the average verification token usage across the five evaluated benchmarks, as summarized in Table 3. In general, the cost of solution verification is lightweight compared to the problem-solving process, with ratios ranging from 0.02 to 0.14. The verification responses optionally highlight the critical issues and provide the final score succinctly. This property allows us to incorporate

Table 3: Average reasoning and verification token usage across RISE models.

| Model | Reason. | Veri. | Ratio (V/R) |
|-------|---------|-------|-------------|
| RISE-1.5B | 676 | 13 | **0.02** |
| RISE-3B | 686 | 94 | **0.14** |
| RISE-7B | 693 | 52 | **0.08** |

the verification outputs from RISE models at test time (e.g., through weighted majority voting) without introducing significant computational overhead, thereby maintaining fairness in comparison to baseline budgets and preserving the robustness of RISE under realistic evaluation settings. We also provide the results with adjusted sampling budgets in Table 4 for reference, where RISE continues to improve upon vanilla majority voting and outperforms the Zero-RL baselines in most cases.

Table 4: Test-time scaling (maj@k) performance under adjusted sampling budgets. Zero-RL and RISE results are measured under baseline budget; SV denotes Self-verify.

| Model | RISE + SV Budget | Baseline Budget | Zero-RL | RISE | RISE + SV |
|---|---|---|---|---|---|
| $k = 4$ | | | | | |
| Qwen2.5-1.5B | 4.08 sols | 4 sols | 26.5 | 28.3 | **28.5** |
| Qwen2.5-3B | 4.52 sols | 5 sols | 37.3 | **38.6** | 38.5 |
| Qwen2.5-7B | 4.32 sols | 4 sols | 45.6 | 46.7 | **48.6** |
| $k = 8$ | | | | | |
| Qwen2.5-1.5B | 8.16 sols | 8 sols | 29.1 | 31.4 | **31.7** |
| Qwen2.5-3B | 9.12 sols | 9 sols | 39.1 | 39.2 | **40.7** |
| Qwen2.5-7B | 8.64 sols | 9 sols | 49.1 | 48.2 | **49.8** |

## G.3 Detailed Analysis for Enhanced Verification

In Figure 7, we report the average verification frequency and accuracy of self-verified solutions on the five benchmarks. Here, we present the fine-grained results between RISE models and Zero-RL baseline on each evaluation benchmark.

Table 5: Performance comparison between RISE models and Zero-RL models on verification frequency and effectiveness for the generation.

| Method | Verification Frequency | | | | | |
|---|---|---|---|---|---|---|
| | MATH | AIME | AMC | Minerva | Olympiad | Avg. |
| Qwen2.5-1.5B-Zero-RL | 6.45 | 6.67 | **7.81** | 2.25 | **15.59** | 7.75 |
| **RISE-1.5B** | **7.10** | **8.75** | 5.31 | **2.53** | 15.31 | **7.80** |
| Qwen2.5-3B-Zero-RL | **4.90** | 8.33 | 14.29 | 2.99 | 8.72 | 7.85 |
| **RISE-3B** | 4.63 | **9.17** | **18.18** | **3.08** | **9.67** | **8.94** |
| Qwen2.5-7B-Zero-RL | 5.30 | 5.00 | 7.19 | 1.56 | 8.19 | 5.45 |
| **RISE-7B** | **6.08** | **7.92** | **8.13** | **1.79** | **8.57** | **6.50** |
| | Self-Verified Reasoning Accuracy | | | | | |
| Qwen2.5-1.5B-Zero-RL | 37.21 | 0.00 | 24.00 | **24.49** | **15.59** | 20.26 |
| **RISE-1.5B** | **38.73** | **4.76** | **35.29** | 23.64 | 15.31 | **23.55** |
| Qwen2.5-3B-Zero-RL | **45.92** | 0.00 | 14.29 | 20.00 | 19.96 | 20.03 |
| **RISE-3B** | 43.78 | **4.55** | **18.18** | **22.39** | 19.54 | **21.69** |
| Qwen2.5-7B-Zero-RL | 62.74 | 0.00 | 8.70 | **35.29** | 26.70 | 26.68 |
| **RISE-7B** | **65.43** | **5.26** | **38.46** | 28.21 | **28.73** | **33.22** |

Table 6: Reflection Keywords Rate between RISE models and Zero-RL models.

| Method | Verification Frequency in Generation | | | | | |
|---|---|---|---|---|---|---|
| | MATH | AIME | AMC | Minerva | Olympiad | Avg. |
| Qwen2.5-1.5B-Zero-RL | 0.16 | 0.40 | 0.26 | **0.16** | 0.29 | 0.25 |
| **RISE-1.5B** | **0.19** | **0.45** | **0.29** | **0.16** | **0.32** | **0.28** |
| Qwen2.5-3B-Zero-RL | 0.14 | 0.40 | **0.24** | 0.11 | 0.27 | 0.23 |
| **RISE-3B** | **0.16** | **0.45** | 0.20 | **0.13** | **0.29** | **0.25** |
| Qwen2.5-7B-Zero-RL | 0.13 | 0.38 | 0.23 | 0.08 | 0.23 | 0.21 |
| **RISE-7B** | **0.14** | **0.50** | **0.29** | **0.10** | **0.27** | **0.26** |

## G.4 Reflection Keywords Analysis

Following Yeo et al. [2025], we track the self-reflection keywords {"wait", "however", "alternatively", "retry", "recheck"} to quantitatively measure the general reflection behaviors beyond the

self-verification among the model problem-solving responses. In practice, we sum the total word counts for the keyword set and normalize it by the dataset size.

The results in Table 6 show that our RISE model constantly have a higher level of reflection-related behaviors than the Zero-RL models, indicating the positive effect of self-verification training.

# H    Generalization Results

## H.1    Generalization on Qwen3 Models

Table 7: Performance comparison of RISE and baseline methods for Qwen3 models.

| Model | *Reasoning* | | | | | | *Self-Verification* | | | | | |
|---|---|---|---|---|---|---|---|---|---|---|---|---|
| | MATH | AIME | AMC | Mine. | Olym. | Avg. | MATH | AIME | AMC | Mine. | Olym. | Avg. |
| *Qwen3-4B-Base* | | | | | | | | | | | | |
| Base | 39.4 | 6.3 | 24.1 | 12.6 | 17.8 | 20.0 | 60.9 | 72.6 | 61.9 | **59.4** | 63.8 | 63.7 |
| Zero-RL | 73.7 | **13.3** | 45.9 | 29.5 | 37.2 | 39.9 | 73.7 | 39.9 | 52.9 | 37.9 | 47.8 | 50.4 |
| **RISE** | **77.8** | 12.9 | **52.8** | **43.4** | **40.6** | **45.5** | **87.4** | **79.6** | **70.9** | 50.8 | **68.0** | **71.3** |
| *Qwen3-8B-Base* | | | | | | | | | | | | |
| Base | 42.5 | 8.3 | 28.4 | 15.4 | 18.4 | 22.6 | 67.0 | 65.5 | 64.4 | **62.9** | 62.0 | 64.4 |
| Zero-RL | 77.6 | 13.8 | 58.1 | 37.7 | 41.6 | 45.7 | 79.7 | 54.1 | 68.8 | 46.9 | 56.9 | 61.3 |
| **RISE** | **83.0** | **21.3** | **59.4** | **48.4** | **44.4** | **51.3** | **91.8** | **85.4** | **87.4** | 53.4 | **72.2** | **78.1** |

To further demonstrate the effectiveness of RISE, we conduct a new set of experiments on the latest Qwen3 models (Apache 2.0 License). As the mathematical reasoning capability of Qwen3 has been shown to be substantially higher than that of Qwen2.5 [Yang et al., 2025], we construct a more challenging training set to better match their ability. Specifically, we downsample the DeepMath-103K [He et al., 2025] dataset (MIT License) by difficulty level to obtain a 10K subset. Using this data, we train both Qwen3-4B-Base and Qwen3-8B-Base models with the Zero-RL (vanilla PPO) baseline and our RISE, under identical configurations except for the inclusion of the verification objective. For RISE models, we set the verification batch size to 128 by default, consistent with the Qwen2.5 experiments. After training, we follow the same evaluation protocols as in the main experiment to assess model performance on both problem-solving and solution verification tasks. The results, summarized in Table 7, show that RISE achieves substantial improvements in reasoning accuracy over the Zero-RL baseline, with average gains of +5.6% and +5.6% for the 4B and 8B models, respectively. Verification accuracy also improves markedly (+20.9% for the 4B model and +16.8% for the 8B model). Together with our main experiments on Qwen2.5, these results demonstrate the generalizability and effectiveness of RISE across different model scales and families.

## H.2    Generalization on Diverse Tasks

Table 8: Performance of RISE and baseline models on diverse tasks.

| Model | MMLU-Pro | GPQA | HumanEval | Veri. Acc. (Avg) |
|---|---|---|---|---|
| Qwen2.5-1.5B-Zero-RL | 19.7 | 20.0 | 39.9 | 21.4 |
| **RISE-1.5B** | **21.3** | **23.0** | **44.7** | **67.7** |
| Qwen2.5-7B-Zero-RL | 46.3 | 27.8 | 63.7 | 49.5 |
| **RISE-7B** | **47.4** | **28.3** | **64.1** | **62.1** |

To evaluate the cross-domain generalization of RISE beyond mathematical reasoning, we further conduct a zero-shot transfer study on RISE models. Concretely, we directly test the math-tuned models on diverse tasks, including general knowledge (MMLU-Pro), science (GPQA-Diamond), and code generation (HumanEval). The results in Table 8 show that RISE-1.5B and RISE-7B consistently outperform their Zero-RL baselines in both reasoning and verification accuracy across these out-of-distribution domains. These findings indicate that the self-verification ability learned by RISE is

robust and transferable, providing benefits even in tasks unseen during training. Future work could try applying RISE to broader domains where verifiable reward signals are available.

### H.3 Comparison to Separate Training of Reasoning and Verification

While our main focus is to improve the reasoning capability of a single LLM rather than building a multi-agent system with specialized components, we provide pilot results on separately fine-tuned solver–verifier models using the medium-sized Qwen2.5-3B. Specifically, we use the same MATH-Hard dataset and finetune an additional verifier model on the Zero-RL model's (i.e., solver) solutions, formatted as verification problems. As shown in Table 9, the results averaged over five benchmarks indicate that RISE achieves higher overall reasoning performance, despite being slightly less verification-accurate than the separately trained verifier. Unlike this separate training scheme, RISE improves model performance holistically and scales more efficiently, requiring significantly less training and inference cost.

Table 9: Reasoning and verification accuracy for Qwen2.5-3B Finetuned Models.

| Model | Reason. | Veri. |
|---|---|---|
| Base | 14.1 | 25.3 |
| Zero-RL (Solver) + Zero-RL (Verifier) | 32.5 | **78.3** |
| RISE | **33.5** | 74.3 |

### H.4 Verification on External Solution Set

Although we mainly use the verification ability of RISE models for self-improvement purposes, we construct a test set with external solutions to address potential concerns about verification generalizability. Specifically, we use GPT-4o's solutions on the five evaluated benchmarks and ask the models to verify their correctness. The results are shown in Table 10, which indicate that RISE models still maintain a high level of verification accuracy on the external solutions. This suggests that the verifier learned by RISE generalizes well beyond its own generation style and has acquired a robust notion of correctness.

Table 10: Verification performance of RISE models and other verifiers on the external solution set.

| Verifier Model | Verification Accuracy |
|---|---|
| RISE-1.5B | 67.9 |
| RISE-3B | 74.4 |
| RISE-7B | 70.7 |
| GPT-4o | 57.8 |
| Math-Shepherd | 58.9 |

## I   Cases Study

---

**Case Study**

### Analysis:
The response provided contains several errors and incorrect calculations. Here's a step-by-step breakdown of the issues:
1. **Triangle Type Identification**: The response incorrectly identifies the triangle as a right triangle based on the Pythagorean theorem, but it does not correctly apply the theorem to confirm the right angle.
2. **Angle Bisector Theorem Application**: The Angle Bisector Theorem is applied incorrectly, leading to an incorrect ratio for $BD$ and $DC$.
3. **Length Calculation of $AD$**: The length of the angle bisector $AD$ is calculated incorrectly using the formula for the length of the angle bisector in a right triangle.
4. **Power of a Point Theorem**: The application of the Power of a Point theorem is incorrect, leading to an incorrect value for $AP$.

### Final Rating:
\\boxed{-0.5}

---

Figure 14: Example verification returned by RISE-7B on AIME 24.

