# OpenReview forum: "Trust, But Verify: A Self-Verification Approach to Reinforcement Learning with Verifiable Rewards"
_NeurIPS.cc/2025/Conference — NeurIPS 2025 poster_

### Official Review · Reviewer_KxXf · 2025-06-04

**Clarity:** 2
**Significance:** 2
**Originality:** 2
**Rating:** 3
**Confidence:** 4

**Summary:**

The paper introduces a new framework for improving mathematical reasoning of language models. The core idea is to train a language model jointly on generation and verification tasks using reinforcement learning with verifiable rewards. The proposed method achieves modest gains on the generation task and substantial gains on the verification task across a range of mathematical reasoning benchmarks.

**Questions:**

Please refer to the second and last bullet points of the weaknesses above.

**Ethical Concerns:**

["NO or VERY MINOR ethics concerns only"]

**Final Justification:**

I initially considered their evaluation unfair and insufficiently robust, but their rebuttal addressed my concerns, so I have raised my score from 2 to 3. I still feel that the results on Qwen2.5 are not strong enough to convincingly demonstrate that the proposed method outperforms the baselines. The Qwen-3 experiments are highly promising, but I excluded them from the evaluation to maintain fairness (GPU-poor vs. GPU-rich).

**Limitations:**

The authors clearly states the limitations of the proposed method in Appendix A.

**Quality:**

2

**Strengths And Weaknesses:**

Strengths
- (Originality) Training a single unified model for generation and verification is novel.
- (Quality) The proposed method is evaluated on various mathematical reasoning benchmarks.

Weaknesses
- (Significance) The improvement in generation accuracy is not statistically significant, except on the AMC dataset.
- (Significance) Verification does not seem to offer much benefit at test time. Because the method uses a generative verifier, each verification step consumes roughly the same number of tokens as producing a solution. For a fair comparison, "k=4 + verification" should be compared against "k=8 without verification." Based on Figure 2, the latter achieves higher accuracy.
- (Clarity) The online and offline verification experiment (Figure 6) shows that high verification accuracy does not necessarily translate into high generation accuracy, thereby unintentionally weakening the paper’s claim.

---

> ### Author Rebuttal · Authors · 2025-07-31
>
> We thank Reviewer KxXf for their critical feedback, which has pushed us to further strengthen our paper with new experiments and analysis.
>
> > **W1:** The improvement in generation accuracy is not statistically significant, except on the AMC dataset.
>
> While the gains with the Qwen2.5 series are consistent, we acknowledge that stronger improvement is helpful. To further demonstrate the effectiveness and significance of RISE, we ran new experiments on the more recent **Qwen3 base models, Qwen3-4B-Base and Qwen3-8B-Base**, using 10K training data from DeepMath.
>
> **On these newer models, RISE achieves substantial and significant improvements in reasoning accuracy over the Zero-RL (PPO) baseline**, with average gains of **+5.6%** and **+5.6%** for the 4B and 8B models, respectively. Verification accuracy also sees a massive boost (+20.9% for the 4B model and +16.8% for the 8B model).
>
> | Reasoning Acc. (**4B**)| MATH | AIME | AMC | Minera | Oly. | **Avg.** |
> | --- | --- | --- | --- | --- | --- | --- |
> | PPO | 73.7 | 13.3 | 45.9 | 29.5 | 37.2 | 39.9 |
> | **RISE** | **77.8** | 12.9 | **52.8** | **43.4** | **40.6** | **45.5 (+5.6)** |
>
> | Reasoning Acc. (**8B**)| MATH | AIME | AMC | Minera | Oly. | **Avg.** |
> | --- | --- | --- | --- | --- | --- | --- |
> | PPO | 77.6 | 13.8 | 58.1 | 37.7 | 41.6 | 45.7 |
> | **RISE** | **83.0** | **21.3** | **59.4** | **48.4** | **44.4** | **51.3 (+5.6)** |
>
> These results clearly demonstrate the significant impact of our method. We will add them to the paper.
>
> ---
>
> > **W2:** Test-time benefit and fair comparison of compute.
>
> This concern is based on a **critical factual misunderstanding about the compute cost of verification.** A verification response can be very short (e.g., "The final rating for the response is \\boxed{-0.5}."), whereas a solution response is a long chain-of-thought. Thus, the premise that their token costs are "roughly the same" is incorrect.
>
> We calculated the actual token consumption. On average, a **verification response uses only 2% to 14% of the tokens of a full solution generation.**
>
> | Model | Avg. Reasoning Tokens | Avg. Verification Tokens | **Ratio (Verif/Reason)** |
> | --- | --- | --- | --- |
> | RISE-1.5B | 676 | 13 | **0.02** |
> | RISE-3B | 686 | 94 | **0.14** |
> | RISE-7B | 693 | 52 | **0.08** |
>
> Therefore, comparing `k=4 + verif` to `k=8` is not a fair comparison. A much fairer comparison, which generously **overestimates** the verification cost, would be against `k=4 + 4*0.14 ≈ k=5` for the RISE models.
>
> Here is the adjusted comparison. **RISE with self-verification at k=4 consistently outperforms the stronger PPO baseline at k=5.** This demonstrates a clear and compute-efficient benefit at test time.
>
> | maj@k | RISE + self-verify (k=4) | PPO (original, k=4) | PPO (adjusted, k=5) |
> | --- | --- | --- | --- |
> | RISE-1.5B | **28.5** | 26.5 | 28.1 |
> | RISE-3B | **38.5** | 36.2 | 37.3 |
> | RISE-7B | **48.6** | 45.6 | 46.9 |
>
> ---
>
> > **W3:** Offline verification experiment weakens the claim.
>
> We believe there is a misunderstanding of the purpose of this ablation. The result in Figure 6 does not weaken our claim; on the contrary, **it strengthens our central thesis that *online* verification is crucial.** The entire point of the offline experiment was to show that simply mixing in offline verification data is *not* effective, as the verification skill is not learned jointly with on-policy generation. This highlights the limitation of decoupled approaches and motivates the need for our integrated, online framework.
>
> Additionally, as discussed in Section 5.5, verification is generally an easier task than reasoning, so the translation from verification accuracy to reasoning accuracy is not always linear. In Figure 7 (right), we show that **the accuracy of self-verified responses improves substantially with RISE**, providing a more direct evidence that strengthened self-verification contributes to better reasoning performance.

---

> ### Comment · Reviewer_KxXf · 2025-08-01
>
> Thank you for the detailed response. I will focus my evaluation on the technical aspects of your rebuttal, but I feel it is necessary to comment on the tone. Phrases such as "This concern is based on a critical factual misunderstanding about the compute cost of verification" and "We believe there is a misunderstanding of the purpose of this ablation" are unnecessarily confrontational. Please see my response below.
>
> **Test-time benefit and fair comparison of compute**
>
> This issue is not due to a misunderstanding. It arises from a **lack of information** in the manuscript. Because a generative verifier necessarily produces a full chain-of-thought, so it is reasonable to expect a substantial token cost. The example in Figure 14 appears to be nearly 100 tokens. If the paper does not account for this compute, I believe pointing it out is a valid critique. Moreover, a recent paper has shown that generative verification is ineffective under the small compute budgets [1], so I believe this concern is well founded. Finally, because my concern here is solely about test-time compute, the appropriate baseline is RISE without verification, not PPO.
>
> **Offline verification experiment weakens the claim**
>
> This issue is not the result of a misunderstanding. Training the verifier using an offline dataset does not improve reasoning accuracy (Figure 6, left).
>
> **Summary**
>
> The paper claims that training a single model as both generator and verifier using RL and using online solutions to generate verifications, improves both generation and verification performance. However, I still have two concerns:
> 1. Whether the verifier is trained on offline or online solutions seems to have no meaningful impact on reasoning accuracy.
> 2. Even when the verifier achieves higher accuracy, it offers no meaningful gain in reasoning performance at test time once the additional compute cost is taken into account.
>
> **References**
>
> [1] Nishad Singhi et al., When To Solve, When To Verify: Compute-Optimal Problem Solving and Generative Verification for LLM Reasoning, arXiv 2025.

---

> ### Author Response · Authors · 2025-08-02
>
> Dear Reviewer KxXf,
>
> Thank you for the detailed follow-up and for pushing us to further clarify our contributions. We sincerely apologize if our previous phrasing came across as confrontational; our intent was only to highlight the key technical distinctions that we believe are central to our work. We value this discussion and have incorporated your feedback to strengthen our analysis.
>
> We will address your two remaining concerns below.
>
> ---
>
> **1. On the Test-Time Benefit and Compute Cost of Self-Verification**
>
> We agree that the token cost of verification was not detailed in the original manuscript, and we thank you for pointing out this omission. We will add a detailed breakdown to the appendix to prevent any confusion.
>
> Your concern is that the compute cost of verification may outweigh its benefits. As you advise, the baseline now changes to the RISE model itself, using majority voting. Our analysis shows that even after accounting for the minor additional cost, self-verification provides the performance gain, especially for larger models and in low-budget scenarios.
>
> **Verification is computationally inexpensive.** As shown in our new analysis (including the latest Qwen3 models), a verification response consumes only a small fraction of the tokens required for a full solution generation.
>
> | Model | Avg. Reasoning Tokens | Avg. Verification Tokens | **Ratio (Verif/Reason)** |
> | --- | --- | --- | --- |
> | RISE-1.5B | 676 | 13 | **2%** |
> | RISE-3B | 686 | 94 | **14%** |
> | RISE-7B | 693 | 52 | **8%** |
> | Qwen3-4B-RISE | 785 | 91 | **12%** |
> | Qwen3-8B-RISE | 1511 | 121 | **8%** |
>
> **Self-verification improves performance at a matched compute budget.** Following your suggestion, we compare `RISE + self-verify` against a `RISE` baseline with an equivalent solution budget. The results show the advantage for using the lightweight self-verification explicitly.
>
> - **Comparison at k=4:** Comparing against the equivalent `RISE` baselines, `RISE + self-verify` shows the benefit on 4 out of 5 models, with the largest models seeing the biggest gains.
>
> | Model | Budget (`k=4 + verif`) | Baseline Budget | RISE maj@k | **RISE + self-verify** |
> | --- | --- | --- | --- | --- |
> | RISE-1.5B | 4.08 sols | 4 | 28.3 | **28.5 (+0.2)** |
> | RISE-3B | 4.52 sols | 5 | **38.6** | 38.5 (-0.1) |
> | Qwen3-4B-RISE | 4.48 sols | 4 | 49.4 | **49.7 (+0.3)** |
> | RISE-7B | 4.32 sols | 4 | 46.7 | **47.8 (+1.1)** |
> | Qwen3-8B-RISE | 4.32 sols | 4 | 53.8 | **55.3 (+1.5)** |
> - **Comparison at k=2:** The benefit is even more pronounced in low-budget scenarios, where each sample is more critical. At k=2, self-verification provides a substantial boost across all model sizes.
>
> | Model | Budget (`k=2 + verif`) | Baseline Budget | RISE maj@k | **RISE + self-verify** |
> | --- | --- | --- | --- | --- |
> | RISE-1.5B | 2.04 sols | 2 | 23.7 | **24.3 (+0.6)** |
> | RISE-3B | 2.28 sols | 2 | 30.5 | **31.8 (+1.3)** |
> | Qwen3-4B-RISE | 2.24 sols | 2 | 46.6 | **47.9 (+1.3)** |
> | RISE-7B | 2.16 sols | 2 | 42.1 | **43.6 (+1.5)** |
> | Qwen3-8B-RISE | 2.16 sols | 2 | 53.9 | **57.3 (+3.4)** |
>
> Finally, we wish to highlight another key result: even **without** explicit test-time verification, RISE-trained models achieve **higher majority voting accuracy than PPO models** under an equivalent budget (see Figure 2). This suggests that our method inherently produces more self-consistent models, offering a built-in test-time advantage.
>
> ---
>
> **2. On the Impact of Online vs. Offline Verification on Reasoning**
>
> We appreciate you highlighting the result from the offline verification experiment. To be precise, offline verification does yield a small, consistent improvement over the PPO baseline across all model scales, as shown in the table below.
>
> | Avg. Reasoning Acc. | Zero-RL (PPO) | w/ Offline Self-Verification |
> | --- | --- | --- |
> | Qwen2.5-1.5B | 24.0 | **24.4 (+0.4)** |
> | Qwen2.5-3B | 32.5 | **33.1 (+0.6)** |
> | Qwen2.5-7B | 41.7 | **42.2 (+0.5)** |
>
> However, you are right that this impact is not large, and we believe this finding is not a *weakness* of our work, but rather a **motivation** for it. The core thesis of our work is that the verification skill must be learned *online*, using the model's own on-policy generations, to meaningfully improve reasoning. The relative ineffectiveness of the offline approach is precisely the evidence that supports our claim and highlights the necessity of our proposed **online** framework, RISE.
>
> In contrast to the small gains from offline data, our online method, RISE, yields higher and more meaningful reasoning improvements. As shown in our first rebuttal, on the newer Qwen3 models, **RISE achieves average gains of** **+5.6% on both the 4B and 8B models over the PPO baseline across 5 benchmarks**. This difference between the offline and online results confirms that the tight, integrated loop in RISE is the key to its success.

---

> ### Author Response · Authors · 2025-08-02
>
> **Summary**
>
> In summary, we address your remaining concerns with new data and clarifications:
>
> > Concern 1: Whether the verifier is trained on offline or online solutions seems to have no meaningful impact on reasoning accuracy.
>
> Our method, RISE, is built on **online** self-verification. Our experiments demonstrate that this online training of verification **consistently and meaningfully improves reasoning accuracy**. On the newer Qwen3 models, RISE delivers average gains of **+5.6%** over the PPO baseline, with improvements of **over +10%** on the Minerva Math. The limited gains from the offline ablation only strengthen our claim that **learning from online, on-policy data is critical**, validating the core design of the RISE framework.
>
> > Concern 2: Even when the verifier achieves higher accuracy, it offers no meaningful gain in reasoning performance at test time once the additional compute cost is taken into account.
>
> Verification responses are **computationally efficient** (2–14% of a solution's token cost). Our new experiments, using the fair comparison you suggested, show that adding self-verification provides a lightweight yet effective performance boost, especially for larger models and in low-budget settings (e.g., **+3.4% at k=2** for the Qwen3-8B model). Furthermore, RISE-trained models exhibit superior majority voting accuracy even *without* explicit verification, demonstrating an inherent test-time advantage.
>
> We hope these clarifications can address your concerns. We thank you again for your constructive engagement.

---

> ### Author Response · Authors · 2025-08-04
>
> Dear Reviewer KxXf,
>
> As the discussion period is concluding, we want to thank you again for your thoughtful engagement. Your feedback encouraged us to demonstrate the effectiveness of our method on more recent models, run compute-matched test-time evaluations, and clarify the motivation behind our online training framework, all of which we believe have meaningfully strengthened the paper.
>
> We hope our detailed responses and the new results have addressed your concerns. We remain available for any further questions you may have.
>
> Best regards,
>
> The Authors

---

> ### Comment · Reviewer_KxXf · 2025-08-05
>
> Thank you for conducting new experiments, which make your claims much stronger. I recommend dropping the left plot of Figure 6. In that plot, Qwen2.5-1.5B-Instruct shows higher reasoning accuracy with offline verification than with online verification, which partially weakens your claim. I believe it is sufficient to claim that training with online verification boosts verification accuracy and allows the model to function reliably as a verifier at test time. My initial lower score reflected concerns about the robustness of your evaluation, but after our discussion, I now find the evaluation much more reliable. Please add the new test-time scaling experiments to the Appendix; including them will clearly make your paper even stronger.

---

> > ### Author Response · Authors · 2025-08-05
> >
> > Thank you for your feedback. As suggested, we will revise Figure 6 and include the new test-time scaling results in the Appendix. We're glad the additional experiments addressed your concerns, and would be grateful if you would consider raising your final rating. We truly appreciate this thoughtful and constructive discussion with you.

---

### Official Review · Reviewer_PrjU · 2025-06-26

**Clarity:** 2
**Significance:** 3
**Originality:** 2
**Rating:** 5
**Confidence:** 3

**Summary:**

The paper proposes RISE, a novel RL framework to fine tune an LLM to both reason and verify its own reasoning steps.

**Questions:**

Why would it be desirable to constrain the generator and the verifier to be the same model?
As far as I know, and as the authors state in line 31, these are two independent tasks which are often performed by separate models.
These models could even be of different sizes, to better adapt to the different complexities of the two tasks.

Line 26-27: I don't understand, why would that be a problem? Again, why is it desirable / important for the generator model to have robust self-assessment skills?
Line 33: it seems it is assumed this is a limitation, but it's never argued why is it so.


I am quite unfamiliar with the related literature and I have little to no expertise on this topic, but I suspect the claim in Sec 2 line 80-82 might require some further clarification.
The authors seem to claim no previous work fine-tuned the verifier model using verifiable rewards, is that correct?


Sec 5.1: do you consider any baseline where generator and verifier are RL-tuned independently? That seems a much more relevant baseline than Zero-RL, which only finetunes the generator/problem-solver. Also, how is SFT trained?

Sec 5.2: lines 208-211: this paragraph seems to suggest the authors expect the verification performance of a model to decrease with model size and that RISE limits this problem.
However, I would expect the verification performance to increase with model size; instead, in Table 1, it seems that the verification performance of RISE decreases.
Still, other models in Table 1 generally show improved verification performance when increasing parameters.
Hence, I have some doubts about the strength of the authors' argument, can they maybe comment on this?


Lines 320-325: the author claims "substantial increase in verification frequency", but that seems a rather modest increase to me (between 0 and 1%?).
Also, I don't see why an increased verification frequency is necessarily a desirable property. I imagine one would like the model to ask for verification only when needed (for example, when it suspects the reasoning trajectory is mistaken and can, therefore, be pruned). A model requiring verification at each single step would achieve maximum verification frequency but seems undesirable to me.

Is it maybe possible to formulate a different and more insightful metric to do this comparison? E.g. can we measure how often both models require verification versus how often verification is actually needed? (e.g. an intersection-over-union metric).
On this note, the following comparison between Zero-RL and RISE in terms of overall reasoning accuracy rather than verification frequency seems much more useful to me.

**Ethical Concerns:**

["NO or VERY MINOR ethics concerns only"]

**Final Justification:**

The authors clarified all my doubts, included some initial promising analysis and promised to try to implement the baseline suggested.

**Limitations:**

LIMITATIONS

Yes, the authors discuss the limitations of the proposed method.


SOCIETAL IMPACT

The authors claim no societal impacts as their work focuses on improving reasoning capabilities on tasks such as mathematics.
However, claiming that improving reasoning capabilities just has no societal impacts seems a bit an oversimplification to me, although, again, this topic is outside my field of expertise and, therefore, I do not have any strong opinion here.

Still, I know at least that some research on this topic does exist (e.g. potential societal impact when these methods are applied outside mathematics data) so I suspect the conclusion might not be as straightforward as presented here.

What do the authors and the other reviewers think of this?

**Quality:**

3

**Strengths And Weaknesses:**

The paper includes an extensive experiment section, with different ablation studies.
The proposed idea seems interesting and probably quite novel as far as I know, although I have very limited expertise and knowledge of the related literature, so I might be missing some relevant previous work.

On the other hand, I feel like the explanation of the motivation behind the main idea is a bit weak (why should the generator and the verifier be the same model?) and that the manuscript contains a few hand-wavy claims which I am not sure are properly supported.
See Questions below.

Finally, I feel like the authors should consider some additional baselines for proper comparison; see questions below.

---

> ### Author Rebuttal · Authors · 2025-07-31
>
> We thank the reviewer PrjU for finding our idea “interesting and probably quite novel”.
>
> > **Q1&Q2:** Motivation for a single, self-verifying model.
>
> These questions get to the heart of our motivation. Our goal is **not just to build a separate verifier, but to improve the generator's own reasoning process.**
>
> The key issue we address is "superficial self-reflection", where the emergent self-reflections do not lead to correct final answers. By training a **single, unified model** to both solve and verify, RISE encourages the model to develop an **internally consistent** **and correct** reasoning process. A model that can reliably check its own work is more likely to produce robust and trustworthy reasoning. This aligns with trends in state-of-the-art models (e.g., OpenAI's o1, DeepSeek-R1) which heavily feature self-reflection as a core capability.
>
> ---
>
> > **Q3:** I am quite unfamiliar with the related literature and I have little to no expertise on this topic, but I suspect the claim in Sec 2 line 80-82 might require some further clarification. The authors seem to claim no previous work fine-tuned the verifier model using verifiable rewards, is that correct?
>
> To clarify our claim in Lines 80-82: our novelty lies in **explicitly and simultaneously training both the generator and the verifier within a single, online RL loop, using on-policy data and a unified objective.** While others have trained verifiers separately or used them to generate rewards, to our knowledge, this tightly-coupled, joint-training framework for both skills is novel.
>
> ---
>
> > **Q4:** Sec 5.1: do you consider any baseline where generator and verifier are RL-tuned independently? That seems a much more relevant baseline than Zero-RL, which only finetunes the generator/problem-solver. Also, how is SFT trained?
>
> Tuning a separate verifier is an interesting direction, but it addresses a different research question (building the best separate components). Our primary goal is to ablate the effect of our proposed joint-training objective. Therefore, the most direct and scientifically sound baseline is **Zero-RL**, which uses the identical RL setup but without our proposed verification objective. This allows for a clean comparison demonstrating the benefit of RISE's core contribution.
>
> Regarding SFT: SFT models are instruction-tuned on the golden solutions from the MATH-Hard (Level 3-5) dataset, and detailed implementations can be found in **Appendix C.**
>
> ---
>
> > **Q5:** Sec 5.2: lines 208-211: this paragraph seems to suggest the authors expect the verification performance of a model to decrease with model size and that RISE limits this problem. However, I would expect the verification performance to increase with model size; instead, in Table 1, it seems that the verification performance of RISE decreases. Still, other models in Table 1 generally show improved verification performance when increasing parameters. Hence, I have some doubts about the strength of the authors' argument, can they maybe comment on this?
>
> Thank you for the opportunity to clarify. We do **not** claim that verification performance should decrease with model size. We believe the self-verification results in Table 1 show variance across scales, not a decreasing trend. This is likely because self-verification on its own outputs is a relatively easy task that even smaller models can learn well, leading to comparable performances.
>
> To further validate our claim, we evaluated the models on a fixed external solutions (GPT-4o responses on the 5 math benchmarks). Here, we see no decreasing trend; rather, all RISE models maintain strong verification accuracy across scales.
>
> | Verifier Model | Verification Acc. on GPT-4o Solutions (Avg.) |
> | --- | --- |
> | RISE-1.5B | 67.9 |
> | RISE-3B | 74.4 |
> | RISE-7B | 70.7 |
> | GPT-4o | 57.8 |
> | Math-Shepherd | 58.9 |
>
> ---
>
> > **Q6 & Q7**: Verification frequency and metrics.
>
> We completely agree that higher frequency is not the goal; **more *effective* verification is**. As we argue in the paper, the increased frequency is meaningful precisely because it leads to better outcomes.
>
> - First, we kindly note that the paper does not claim a "substantial increase" for all models; we explicitly state the increase for the 1.5B model is modest and the increase "becomes substantial as scale grows (+1.09% for 3B and +1.05% for 7B)" (Line 322)." We will rephrase this in the revised version to ensure clarity.
> - More importantly, as shown in **Figure 7 (right)**, the **accuracy of the self-verified subset of responses is significantly higher for RISE models**. This shows the verifications are not just more frequent, but more *impactful*, helping the model correct or confirm its reasoning path.
> - Regarding new metrics (Q7), we agree that measuring when verification is "needed" would be insightful. However, this is non-trivial to define objectively, as we cannot know the model's internal state of uncertainty. We believe our current metric—showing improved accuracy when self-verification is invoked—is also a direct indicator of its effectiveness.
>
> ---
>
> > **Societal Impact:** The authors claim no societal impacts as their work focuses on improving reasoning capabilities on tasks such as mathematics. However, claiming that improving reasoning capabilities just has no societal impacts seems a bit an oversimplification to me.
>
> Thank you for this important point. You are right that "no societal impact" is an oversimplification. We will revise this section. Our intended meaning was that our work, as a foundational training framework evaluated on public math datasets, does not introduce *immediate, direct* societal risks. However, we acknowledge that any advance in AI reasoning capabilities has potential downstream impacts, both positive and negative. We will add a sentence stating that the method we propose should be developed and applied responsibly.

---

> > ### Comment · Reviewer_PrjU · 2025-08-06
> >
> > Thanks for the detailed answers.
> > While the authors' response answered many of my questions, I still have some doubts.
> >
> > > Tuning a separate verifier is an interesting direction, but it addresses a different research question (building the best separate components). Our primary goal is to ablate the effect of our proposed joint-training objective. Therefore, the most direct and scientifically sound baseline is Zero-RL, [...]
> >
> > I am not fully convinced by this claim. My impression is that an important part of the contributions is having the generator verify itself, rather than having a separate verifier. In that sense, I feel like a good baseline to include is an identical setting where the verifier is not constrained to be the generator itself; hence, my original question about having a verifier RL-tuned simultaneously.
> > In other words, I agree the Zero-RL baseline is a good baseline, but I don't see why you should not ablate the idea of self-verification itself by RL-tuning the verifier too. As far as I know, there any recent works which study better generator-verifier systems, which makes me believe the self-verification design is not yet the obvious choice.
> >
> > Do the author disagree with this conclusion? Again, since I have little expertise in this field, I might not be sufficiently familiar with relevant works which prove me wrong on the benefits of self-verification.
> >
> > > First, we kindly note that the paper does not claim a "substantial increase" for all models; we explicitly state the increase for the 1.5B model is modest and the increase "becomes substantial as scale grows (+1.09% for 3B and +1.05% for 7B)" (Line 322)." We will rephrase this in the revised version to ensure clarity.
> >
> > What I meant in my original question was that 1% didn't seem a substantial increase for the 3B and 7B models. Anyways, I feel like the takeaway is more the fact that verification frequency is not an interesting metric here (rather than effective verification, as the authors state too).
> >
> > Similarly, I am not fully convinced verification accuracy alone gives a full picture either. For example, a model could tweak this metric by choosing to verify only when it knows the solution is already correct.
> > As I initially suggested, it seems more interesting to measure how often the model asks for verification when it is actually needed (e.g., when the reasoning trajectory is indeed mistaken and should, therefore, be pruned).
> >
> > > Regarding new metrics (Q7), we agree that measuring when verification is "needed" would be insightful. However, this is non-trivial to define objectively, as we cannot know the model's internal state of uncertainty. We believe our current metric—showing improved accuracy when self-verification is invoked—is also a direct indicator of its effectiveness.
> >
> > Would it be possible to check the intersection between the verification requests and the mistaken trajectories, maybe on some small or synthetic dataset?
> > In any case, I understand this might not be straightforward to implement in the short time of the rebuttal; if improving these metrics is not feasible, I feel like the authors could at least discuss the limitation of the metrics adopted more explicitly.
> >
> > A part from that, I want to clarify that I agree the overall improved reasoning accuracy (e.g. shown in Fig.7) is already a good indicator of the benefits of the proposed method. My critique is that the additional metrics in page 9 (verification frequency and verification accuracy) do not really provide more insights into how RISE and self-verification help with reasoning, and they risk to be misleading.

---

> ### Author Response · Authors · 2025-08-07
>
> Thank you for your response. We are glad that our initial rebuttal resolved some of your concerns and are willing to discuss the remaining questions you have.
>
> > **On the potential baseline for a separately trained verifier**
> >
>
> In this work, we focus on improving the intrinsic reasoning capability of an LLM by enhancing its self-verification behavior, rather than optimizing the performance of a generator-verifier system. As suggested in prior work [1][2][3], self-verification is considered an essential behavior of strong reasoning models, as it enables the model to assess its own reasoning process and correct any mistakes. Moreover, as noted by [4], a potential benefit of self-verification over using a separate verifier is that reasoning models internally encode a notion of correctness, which self-verification can leverage to further improve verification efficiency.
>
> On the other hand, we agree that verification can be performed by a separate verifier model or even by external tools. However, such setups involve multiple entities and require systematic collaboration between the generator and verifier, which is somewhat beyond the scope of our main focus. In addition, involving a separate verifier model introduces challenges in ensuring a fair comparison to our current experiment setting, due to efficiency differences (e.g., the need to run inference over multiple models).
>
> That said, we believe including this ablation would enrich the research depth of the paper, particularly in evaluating the effectiveness of different system designs. We *will* incorporate an experiment involving separately RL-tuned generator and verifier models and provide the corresponding analysis in the future version, given the limited time and resources during this rebuttal period.
>
> [1] Gandhi, Kanishk, et al. "Cognitive behaviors that enable self-improving reasoners, or, four habits of highly effective stars." *arXiv preprint arXiv:2503.01307* (2025).
>
> [2] Zeng, Weihao, et al. "Simplerl-zoo: Investigating and taming zero reinforcement learning for open base models in the wild." *arXiv preprint arXiv:2503.18892* (2025).
>
> [3] Yang, Shiming, et al. "Demystifying Long Chain-of-Thought Reasoning." *Forty-second International Conference on Machine Learning*.
>
> [4] Zhang, Anqi, et al. "Reasoning Models Know When They're Right: Probing Hidden States for Self-Verification." *arXiv preprint arXiv:2504.05419* (2025).
>
> > **On the limitations for verification frequency and verified-response accuracy metrics;  suggestions for a new metric**
> >
>
> To characterize the reflective behavior of LLMs, prior work has introduced metrics such as verification frequency [1][2], where an increase is often interpreted as a desirable signal due to its empirical correlation with improved reasoning accuracy. However, as previously discussed, this metric has some limitations and should be interpreted with caution. Similarly, we agree that verified-response accuracy (referred to as "verification accuracy" in your response) does not provide the full picture, as it reflects correlation rather than causality or necessity. (That said, we still believe defining the necessity of self-verification is challenging, as verification on correct reasoning paths may still yield benefits, such as reinforcing internal consistency and increasing confidence.) In our revision, we will provide clearer explanations of these metrics' limitations and interpretive boundaries.
>
> To address this concern, we followed your suggestion to examine the intersection between verification requests and mistaken reasoning trajectories. Due to limited time and resources, we conducted this experiment on the **AIME 2024** dataset (30 problems and 240 responses per group). Specifically, we constructed a judging prompt and leveraged an advanced model, GPT-4o, to assess whether each reasoning trajectory contained an error *prior* to the invocation of self-verification. We then reported the percentage of responses that invoke self-verification on erroneous trajectories (i.e., when verification is actually needed), alongside the overall benchmark accuracy.
>
> | **Model** | **Verif. on Mistake (%)** | **Accuracy (%)** |
> | --- | --- | --- |
> | Qwen2.5-1.5B-Zero-RL | 6.7 | 2.1 |
> | **RISE-1.5B** | **8.8 (+2.1)** | **2.9 (+0.8)** |
> | Qwen2.5-3B-Zero-RL | 7.9 | 6.7 |
> | **RISE-3B** | **9.2 (+1.3)** | **7.9 (+1.2)** |
> | Qwen2.5-7B-Zero-RL | 4.6 | 12.1 |
> | **RISE-7B** | **7.5 (+2.9)** | **12.5 (+0.4)** |
> | Qwen3-8B-Zero-RL | 6.3 | 13.8 |
> | **Qwen3-8B-RISE** | **10.0 (+3.7)** | **21.3 (+7.5)** |
>
> The results show that RISE models request verification more often on trajectories that are actually going wrong, and they constantly achieve better overall accuracy. We will conduct further experiments to ensure the reliability of this conclusion.
>
> We hope these clarifications, new results, and our commitment address your concerns. We are grateful for your insightful feedback, which has genuinely helped us strengthen the paper.

---

> > ### Comment · Reviewer_PrjU · 2025-08-08
> >
> > Thanks for addressing my last doubts and for including the new analysis in such a short time.
> >
> > I am satisfied by the authors' answers and will update my score accordingly.

---

> > > ### Author Response · Authors · 2025-08-09
> > >
> > > Thank you for your final feedback and for updating your score. We greatly appreciate your constructive engagement throughout the discussion, and we will incorporate the outcomes of this discussion into the final version of our paper.

---

### Official Review · Reviewer_RZbs · 2025-07-02

**Clarity:** 3
**Significance:** 4
**Originality:** 3
**Rating:** 5
**Confidence:** 5

**Summary:**

This work presents RISE, an RL framework for LLMs that enhances RLVR by jointly learning self-verification as an additional verifiable task. This framing allows for a powerful auxiliary task that augments RLVR techniques with an additional task. They then show the efficacy of RISE through extensive experimentation and ablations to show the model improving both self-verification and task performance. They show that learning promotes both task capabilities and self-verification performance which also leads to improved self-consistency based test-time compute strategies. Finally they ablate the online nature of the learning demonstrating that learning both capabillities online is critical to the augmented performance.

**Questions:**

1. No big questions other than weaknesses section. More out of curiosity, have the authors tried testing RISE on other verifiable domains such as code?
2. How would the authors consider applying a similar strategy to learned reward models?
3. Related to question 1, what would the authors think would be the effects of doing multi-task RLVR with this auxiliary loss?
4. Finally, how sensitive is the performance of this method to the prompt chosen for the verification task?

**Ethical Concerns:**

["NO or VERY MINOR ethics concerns only"]

**Final Justification:**

This paper presents a method in a growing body of work augmenting RLVR with other verifiable auxiliary tasks. Their implementation is simple and would scale well across verifiable domains. They further showed the robustness of this methodology to prompts during the rebuttal.

**Limitations:**

Yes

**Paper Formatting Concerns:**

No concerns

**Quality:**

4

**Strengths And Weaknesses:**

Strengths:
- Thorough experimentation. Particularly want to highlight the ablation of adding in offline self-verification and showed the importance of online verification
- Convincing baselines and experiment design to answer the question of "does self-verification help"
- Showed further utility of learning self-verification through test-time compute experiments.

Weaknesses:
- Small clarity comments
1. perhaps the baseline shouldn't be called zero rl but instead PPO.
2. For section 4.1, could the authors more explicitly detail the verficiation reward?

---

> ### Author Rebuttal · Authors · 2025-07-31
>
> Thank you for the positive evaluation and for recognizing the thoroughness of our experiments and the utility of our method.
>
> > **W1&W2:** Clarity on baseline name and verification reward.
>
> Thank you for these excellent suggestions for improving clarity.
>
> 1. **Baseline Name:** We agree. "PPO" is a more precise and standard name for the baseline. We will change "Zero-RL" to "PPO" in the revised paper.
> 2. **Verification Reward:** Absolutely, let us further clarify it. The verification reward is computed based on the exact match between the score extracted from the model’s verification response and the reward provided by the rule-based outcome verifier for that solution. Specifically:
>   - **+1.0 reward:** for correctly predicting the score in the specified format (e.g., `\\boxed{1}`).
>   - **-0.5 reward:** for predicting an incorrect score in the correct format.
>   - **-1.0 reward:** for generating an invalid format (e.g., missing the `\\boxed{}` wrapper).
>
> We will add these explicit details to Section 4.1.
>
> ---
>
> > **Q1:** More out of curiosity, have the authors tried testing RISE on other verifiable domains such as code?
>
> This is an excellent point. While our math-trained models were not exposed to other domains, we conducted a zero-shot transfer evaluation to test their generalizability. We evaluated the models on general knowledge (MMLU-Pro), science (GPQA), and code (HumanEval).
>
> **The results show that RISE, trained only on math, consistently improves both reasoning and verification accuracy across these diverse, out-of-distribution domains.** This demonstrates that the underlying self-verification skill learned by RISE is robust and transferable.
>
> |  | MMLU-Pro (General) | GPQA (Science) | HumanEval (Code) | Averaged Verification Accuray |
> | --- | --- | --- | --- | --- |
> | Qwen2.5-1.5B-PPO (Zero-RL) | 19.7 | 20.0 | 39.9 | 21.4 |
> | RISE-1.5B | **21.3 (+1.6)** | **23.0 (+3.0)** | **44.7 (+4.8)** | **67.7 (+46.3)** |
> | Qwen2.5-7B-PPO (Zero-RL) | 46.3 | 27.8 | 63.7 | 49.5 |
> | RISE-7B | **47.4 (+1.1)** | **28.3 (+0.5)** | **64.1 (+0.4)** | **62.1 (+12.6)** |
>
> Due to resource constraints, we have not yet conducted full RL training on non-math domains, but we believe these results strongly support our generality claims and we will add them to the paper.
>
> ---
>
> > **Q2:** How would the authors consider applying a similar strategy to learned reward models?
>
> This is an insightful direction. In this work, we pair a more challenging task (reasoning) with a relatively easier auxiliary task (verification), and find that co-training improves both. Reversing this setup, i.e., using generative reasoning to facilitate training a learned reward model, may be less straightforward, as the auxiliary task is harder than the main on. However, it remains a promising direction worth exploring, particularly for the learned generative reward models.
>
> ---
>
> > **Q3:** Related to question 1, what would the authors think would be the effects of doing multi-task RLVR with this auxiliary loss?
>
> We believe RISE is extendable to multi-task RLVR settings. The key would be to ensure that the tasks (and their verification counterparts) are balanced in difficulty and that their reward scales are properly aligned. With a well-curated multi-task dataset, we expect RISE to be highly effective.
>
> ---
>
> > **Q4:** Finally, how sensitive is the performance of this method to the prompt chosen for the verification task?
>
> This is an important practical question. We ran an additional experiment on Qwen2.5-7B using a paraphrased verification prompt with minor structural changes.
>
> **The results show that RISE is robust to the paraphrasing of the verification prompt.** While verification accuracy saw a minor drop, the primary reasoning performance remained stable and still significantly outperformed the PPO baseline.
>
> | Qwen2.5-7B | Reasoning | Self-Verification |
> | --- | --- | --- |
> | RISE (Original Prompt) | 42.9 | 69.2 |
> | **RISE (Modified Prompt)** | **42.5** | **65.4** |
> | PPO Baseline | 41.7 | 46.6 |

---

> ### Comment · Reviewer_RZbs · 2025-08-04
>
> Thank you for the response. It's great to see the followup results and addresses most of the additional questions that I had. I will keep my score to reflect this.

---

> > ### Author Response · Authors · 2025-08-05
> >
> > Dear Reviewer RZbs,
> >
> > Thank you for your positive feedback and for engaging with our responses. We are very glad that the additional results helped clarify the points you raised. We appreciate your time and constructive feedback throughout this process and will incorporate all the discussed changes and new results into the final version.

---

### Official Review · Reviewer_rqQc · 2025-07-05

**Clarity:** 3
**Significance:** 3
**Originality:** 3
**Rating:** 4
**Confidence:** 4

**Summary:**

The work introduces RISE, an online reinforcement-learning framework that jointly optimises an LLM’s problem-solving and self-verification skills. In each PPO iteration the policy first generates chain-of-thought solutions, then critiques its own on-policy answers through a templated verification prompt; both trajectories receive binary verifiable rewards from an outcome checker and are mixed in a single RL objective. Experiments on five mathematics benchmarks with Qwen-2.5 models how that RISE raises reasoning accuracy over a Zero-RL baseline and boosts verification accuracy, the learned verifier even outperforms off-the-shelf discriminative and generative judges.

**Questions:**

Discussed in Weakness.

**Ethical Concerns:**

["NO or VERY MINOR ethics concerns only"]

**Final Justification:**

I have checked reviewers-authors discussion, most of my concerns has been addressed. I will keep my score.

**Limitations:**

Yes

**Quality:**

3

**Strengths And Weaknesses:**

# Strengths
1. The work pinpoints “superficial self-reflection” in RLVR and proposes an elegant, integrated solution. This is simpler than multi-stage or reward-model pipelines.
2. Table 1 shows that RISE improves both reasoning and verification on different sized models and multiple datasets, demonstrating robustness to model size and task difficulty.
3. The learned verifier surpasses GPT-4o and Math-Shepherd in judging correctness, suggesting that in-distribution, joint training can yield a  accurate critic and useful for low-latency deployment.
4. The author also discusses the frequency and efficacy of self-verification,  and qualitative cases reveal that RISE encourages multi-step consistency checks rather than re-statements.

# Weaknesses

1. All experiments are in maths with exact-match verifiers. It is unclear whether the method extends to domains where verification is noisy or differen domain (e.g., code-related QA, general reasoning, factual QA). A transfer study or cross-domain demonstration would strengthen generality claims.
2. While verification accuracy soars, reasoning improvements over Zero - RL are relatively small (≤ 2 pp on most benchmarks). The paper could discuss why better self - checking does not translate into larger solving gains. Also, is there any quantified correlation between the verification accuracy and the final RL performance?
3. Because the verifier learns from on-policy data and shares parameters with the generator, it may overfit to its own solution style. Evaluating on external solutions or holding out a verifier dev set would alleviate this concern.
4. The paper does not address cases where the verifier is confidently wrong, or how such failures might compound during training.

---

> ### Author Rebuttal · Authors · 2025-07-31
>
> We thank Reviewer rqQc for the positive and constructive feedback, highlighting our elegant solution and robust results.
>
> > **W1:** All experiments are in maths with exact-match verifiers. It is unclear whether the method extends to domains where verification is noisy or differen domain (e.g., code-related QA, general reasoning, factual QA). A transfer study or cross-domain demonstration would strengthen generality claims.
>
> This is an excellent point. While our math-trained models were not exposed to other domains, we conducted a zero-shot transfer evaluation to test their generalizability. We evaluated the models on general knowledge (MMLU-Pro), science (GPQA), and code (HumanEval).
>
> **The results show that RISE, trained only on math, consistently improves both reasoning and verification accuracy across these diverse, out-of-distribution domains.** This demonstrates that the underlying self-verification skill learned by RISE is robust and transferable.
>
> |  | MMLU-Pro (General) | GPQA (Science) | HumanEval (Code) | Averaged Verification Accuracy |
> | --- | --- | --- | --- | --- |
> | Qwen2.5-1.5B-PPO (Zero-RL) | 19.7 | 20.0 | 39.9 | 21.4 |
> | RISE-1.5B | **21.3 (+1.6)** | **23.0 (+3.0)** | **44.7 (+4.8)** | **67.7 (+46.3)** |
> | Qwen2.5-7B-PPO (Zero-RL) | 46.3 | 27.8 | 63.7 | 49.5 |
> | RISE-7B | **47.4 (+1.1)** | **28.3 (+0.5)** | **64.1 (+0.4)** | **62.1 (+12.6)** |
>
> Due to resource constraints, we have not yet conducted full RL training on non-math domains, but we believe these results strongly support our generality claims and we will add them to the paper.
>
> ---
>
> > **W2:** While verification accuracy soars, reasoning improvements over Zero - RL are relatively small (≤ 2 pp on most benchmarks). The paper could discuss why better self - checking does not translate into larger solving gains. Also, is there any quantified correlation between the verification accuracy and the final RL performance?
>
> Thank you for this thoughtful question. The difference in gains stems from the fact that verification is an easier task than multi-step reasoning. As noted in prior work, a "Generation-Verification Gap" exists where models learn to verify more easily than they learn to generate correct solutions. Our results are consistent with this phenomenon.
>
> While verification gains don't translate 1-to-1 to reasoning gains, **stronger verification directly contributes to better reasoning**. As we show in **Section 5.5 (Figure 7 right)**, the accuracy of solutions that invoke self-verification is significantly higher in RISE models than in the baseline. This shows that the learned skill is being used effectively.
>
> Regarding a quantified correlation, we show a positive trend between the two reward types during training in **Figure 4**. However, the exact correlation is complex and depends on the base model and dataset. The key takeaway is that improved self-verification, enabled by RISE, provides a more reliable mechanism for the model to identify and correct error reasoning paths, leading to overall performance improvements.
>
> ---
>
> > **W3:** Because the verifier learns from on-policy data and shares parameters with the generator, it may overfit to its own solution style. Evaluating on external solutions or holding out a verifier dev set would alleviate this concern.
>
> This is a valid concern. To test for overfitting, we evaluated our RISE-trained verifiers on their ability to judge the correctness of **external solutions generated by GPT-4o**.
>
> **Our results show that RISE models outperform strong, dedicated verifiers like GPT-4o and Math-Shepherd on these external solutions.** This indicates that the verifier learned by RISE generalizes well beyond its own generation style and has acquired a robust, non-overfit notion of correctness.
>
> | Verifier Model | Verification Acc. on GPT-4o Solutions (Avg.) |
> | --- | --- |
> | **RISE-1.5B** | 67.9 |
> | **RISE-3B** | **74.4** |
> | **RISE-7B** | 70.7 |
> | GPT-4o (Verifier) | 57.8 |
> | Math-Shepherd | 58.9 |
>
> ---
>
> > **W4:** The paper does not address cases where the verifier is confidently wrong, or how such failures might compound during training.
>
> This is handled directly by our reward mechanism. As described in Lines 153-163, the verifier receives a reward signal based on whether its predicted score (e.g., correct/incorrect) matches the ground-truth score from the outcome verifier.
>
> **Critically, if the verifier is confidently wrong (e.g., it labels a wrong solution as correct), it receives a negative reward and is penalized.** This explicit feedback loop directly discourages overconfident errors and prevents them from compounding during training, forcing the verifier’s prediction to align with the ground-truth outcome.

---

> > ### Comment · Reviewer_rqQc · 2025-08-07
> >
> > Thanks for the response, the rebuttal has addressed my concern. I would like to keep my positive score. Hope the detailed discussion will be included in the final version.

---

> > > ### Author Response · Authors · 2025-08-07
> > >
> > > Dear Reviewer rqQc,
> > >
> > > We are glad that our rebuttal has addressed your concerns. We appreciate your time and valuable feedback throughout this process and will incorporate all the discussed changes and new results into the final version.

---

### Comment · Area_Chair_bYMP · 2025-08-03
**Discussion Phase**

Dear Authors and Reviewers,

I would like to thank the authors for providing detailed rebuttal messages. I would also like to thank reviewer KxXf for already engaging in further discussion.

For the other reviewers, I would like to encourage you to carefully read all other reviews and the author responses and engage in an open exchange with the authors. Please post your first response as soon as possible within the discussion time window, so there is time for back and forth discussion with the authors. Ideally, all reviewers will respond to the authors, so that the authors know their rebuttal has been read.

Best regards,
AC

---

### Decision · Program_Chairs · 2025-09-17

**Decision:**

Accept (poster)

**Comment:**

This paper proposes RISE, a new online RL framework for LLMs that learns self-verification as an additional verifiable task, together with the original problem-solving. Initially, the reviewers raised some concerns such as the demonstration on diverse tasks (rqQc, RZbs, KxXf) or the fairness due to the cost from verification step (KxXf). During the rebuttal, the authors successfully resolved these concerns as denoted by the reviewer’s replies. Therefore, we would like to suggest accept this work.